

# Processes culminating in the 2015 phreatic explosion at Lascar volcano, Chile, monitored by multiparametric data

Ayleen Gaete[1], Thomas R. Walter[1], Stefan Bredemeyer[1,2], Martin Zimmer[1], Christian Kujawa[1], Luis Franco[3], Juan San Martin[4], Claudia Bucarey Parra[3]

[1] GFZ German Research Centre for Geosciences, Telegrafenberg, 14473 Potsdam, Germany
[2] GEOMAR Helmholtz Centre for Ocean Research Kiel, 24148 Kiel, Germany
[3] Observatorio Volcanológico de Los Andes del Sur (OVDAS), Servicio Nacional de Geología y Minería (SERNAGEOMIN), Temuco, Chile.
[4] Physics Science Department, Universidad de la Frontera, Casilla 54-D, Temuco, Chile.

*Correspondence to*: Ayleen Gaete (agaete@gfz-potsdam.de)

**Abstract.** Small steam-driven volcanic explosions are common at volcanoes worldwide but are rarely documented or monitored; therefore, these events still put residents and tourists at risk every year. Steam-driven explosions also occur frequently (once every 2-5 years on average) at Lascar volcano, Chile, where they are often spontaneous and lack any identifiable precursor activity. Here, for the first time at Lascar, we describe the processes culminating in such a sudden volcanic explosion that occurred on October 30, 2015, which was thoroughly monitored by cameras, a seismic network, and gas ($SO_2$ and $CO_2$) and temperature sensors.

Prior to the eruption, we retrospectively identified unrest manifesting as a gradual increase in the number of long-period (LP) seismic events in 2014, indicating an augmented level of activity at the volcano. Additionally, $SO_2$ flux and thermal anomalies were detected before the eruption. Then, our weather station reported a precipitation event, followed by changes in the brightness of the permanent volcanic plume and (10 days later) by the sudden volcanic explosion. The multidisciplinary data exhibited short-term variations associated with the explosion, including (1) an abrupt eruption onset that was seismically identified in the 1-10 Hz frequency band, (2) the detection of a 1.7 km high white-grey eruption column in camera images, and (3) a pronounced spike in sulfur dioxide ($SO_2$) emission rates reaching 55 kg sec-1 during the main pulse of the eruption as measured by a mini-DOAS scanner. Continuous $CO_2$ gas and temperature measurements conducted at a fumarole on the southern rim of the Lascar crater revealed a pronounced change in the trend of the relationship between the carbon dioxide ($CO_2$) mixing ratio and the gas outlet temperature; we believe that this change was associated with the prior precipitation event. An increased thermal anomaly inside the active crater observed through Sentinel-2 images and drone overflights performed after the steam-driven explosion revealed the presence of a fracture ~50 metres in diameter truncating the dome and located deep inside the active crater, which coincides well with the location of the thermal anomaly. Altogether, these observations lead us to infer that a lava dome was present and subjected to cooling and inhibited degassing. We conjecture that a precipitation event led to the short-term build-up of pressure inside the shallow dome that eventually triggered a vent-clearing phreatic explosion. This study shows the chronology of events culminating in a steam-driven explosion but also demonstrates that phreatic explosions are difficult to forecast, even if the volcano is thoroughly monitored; these findings also emphasize why ascending to the summits of Lascar and similar volcanoes is hazardous, particularly after considerable rainfall.



## 1 Introduction

Volcanoes possessing an identified region of shallow magma storage often allow the close monitoring of changes associated with deformation, seismicity and degassing activity, all of which are highly beneficial for eruption forecasting or early warning systems (Sparks, 2003). Nevertheless, many volcanic eruptions still occur without clear precursors, highlighting the need to further investigate such types of volcanoes. For instance, unexpected and sudden phreatic eruptions occurred in 1979 at the Diëng Plateau (Le Guern et al., 1982) and in 2014 at Mt. Ontake (Oikawa et al., 2016), killing 142 and 64 people, respectively. Approximately 85% of the phreatic explosions that occur worldwide are not followed by the extrusion of magma (Barberi et al., 1992). Thus, the occurrence of such surprising eruptions implies that the responsible processes and their scales are not easily detectable by conventional geophysical and geochemical instrumentation (Barberi et al., 1992). Volcano monitoring networks are commonly designed to detect precursor activity in the crust, where the movement of magma at depth causes detectable gravity changes, seismicity, deformation or degassing. Accordingly, eruptions that are associated with very shallow processes, such as phreatic explosions, are naturally difficult to monitor beforehand, as shallow water steam explosions generally occur spontaneously, as was identified during the phreatic eruptions at Bandai volcano (Yamamoto et al., 1999), Mt. Ruapehu (Christenson et al., 2010), Mayon volcano (Newhall et al., 2001), Aso volcano (Kawakatsu et al., 2000) and elsewhere (Barberi et al., 1992; Mastin, 1995). Unfortunately, most of the aforementioned examples commonly exhibit a steep morphology, a high altitude and/or hazardous access conditions, and hence, the implementation of conventional monitoring close to their craters is a challenging task.

Possible precursory signs of phreatic eruptions, if identified, generally occur at short notice and may manifest as seismic tremors (Martinelli, 1990), very long-period (VLP) events (Jolly et al., 2010), inflation (Nakamichi et al., 2009), or changes in the gas chemistry and temperature of fumarolic emissions (de Moor et al., 2016) only minutes, hours or days prior to the eruption. The detection of such sudden changes requires monitoring by means of an instrument network close to the summit that combines different geophysical and geochemical techniques (Scarpa and Tilling, 2012).

Lascar, Chile (23° 22' S, 67° 44' W, 5590 m), is a volcano with relatively easy access that has repeatedly exhibited steam-driven explosions. The explosions at Lascar have scarcely been studied, and therefore, little knowledge exists regarding whether these events have been characterized by precursor activity. In an effort to better comprehend this hazardous type of eruption, a monitoring network was installed in 2010. Indeed, a well-monitored volcanic explosion suddenly occurred in 2015, and this event shall be discussed in detail in this paper.

## 2 Study area and explosive history of Lascar volcano

Lascar volcano, the most active volcano in northern Chile and within the Andean Central Volcanic Zone (Francis and Rothery, 1987; Gardeweg et al., 1998; Tassi et al., 2009), formed in association with the subduction of the Nazca plate beneath the South American plate (Jordan et al., 1983). This volcano is located in the region of Antofagasta at distances of approximately 17 and 34 km from the small towns of Talabre and Toconao, respectively, and 68 km from the touristic city of San Pedro de





Atacama. Lascar is an andesitic-dacitic stratovolcano that consists of two overlapping cones, which formed along an ENE-WSW-oriented lineament, and hosts five summit craters (Gardeweg et al., 1998) that partially overlap and interact (de Zeeuw-van Dalfsen et al., 2017). The eruptive behaviour of Lascar exhibits many characteristics, including the continuous emission
of volcanic gases, smaller steam-driven explosions, the generation of thick lava flows and the expulsion of major plinian ash-loaded clouds during the climax of the eruption (Casertano and Barozzi, 1961).

The historical record of Lascar's activity comprises approximately 27 eruptions, almost half of which have occurred in the springtime (mid-September to mid-December), depicting a possible seasonal predilection (see Table 1). The eruptions range from effusive to explosive events, often with phreatic characteristics and the ejection of gas, ash and debris kilometres high
into the atmosphere. Additionally, dome building and collapse have been observed (for further details, see Table 1). The most recent large eruption, classified as having a volcanic explosivity index (VEI) of 4, occurred in January 1993 (González-Ferrán, 1995; Siebert et al., 2010). The 1993 eruption was identified as the climax of a longer episode characterized by 4 cycles of activity initiating in 1984; each cycle was separated by explosive vulcanian eruptions that commenced with minor phreatic activity until the cycle eventually culminated in a large explosion that ejected incandescent material into the air and produced
plinian to subplinian eruption clouds (Matthews et al., 1997). By 1992, a lava dome was identified inside the active crater of Lascar; after the 1993 plinian eruption, a new dome grew inside the crater with an andesitic-dacitic composition (González-Ferrán, 1995). Smaller phreatic and vulcanian eruptions followed in December 1993 and February 1994. The short-lived explosive eruption in December 1993 was accompanied by seismic activity with an intensity of 3 on the modified Mercalli intensity scale (Global Volcanism Program, 1994). In contrast, the seismic activity associated with the other eruptions was
only scarcely felt in the nearby village of Talabre at a distance of 10 km from Lascar (Global Volcanism Program, 1994), implying only low-magnitude seismicity. Following the 1993 eruptive cycle, historical reports document explosions every 2-3 years on average (Table 1) with a frequent occurrence of steam-driven explosions. This kind of explosion commonly occurs without any evidence of precursor activity (e.g., seismicity), due to either the lack of precursors or the lack of a sufficiently dense deployment of instrumentation.

This type of eruption may be poorly understood for numerous potential reasons; among them, prior to 2010, ground-based monitoring studies were conducted only sporadically at Lascar and were restricted to only short-term (weeks to months) campaign-based fieldwork. Therefore, the preparation phases and periods preceding steam-driven explosions were not documented with adequate instrumentation. Remote sensing, on the other hand, could provide evidence distinguishing some unrest periods preceding notable events, such as the 2000 eruption, which was associated with a short-term thermal reduction
and a change in the dimensions of the dome area (Wooster, 2001). Similarly, satellite-based observations were employed to assess the dispersion of eruption products during the 2003 and 2005 unrest periods (Aguilera et al., 2006; Viramonte et al., 2006); unfortunately, these observations lacked the ability to identify precursors. Satellite interferometric synthetic aperture radar (InSAR) data have been used to search for a possible shallow magma plumbing system to explain these steam-driven eruptions, and cooling and contraction processes were revealed accordingly; however, evidence for a magma chamber could
not be provided (Pavez et al., 2006; Pritchard and Simons, 2002).





Despite the difficulty of forecasting steam-driven explosions, some studies have retrospectively identified unexpected precursor activity associated with Lascar's episodes of unrest. Wooster (2001) described the new rapid cooling behaviour of the Lascar dome as precursor activity following the 1993 eruption, while a recent study of the 2013 explosion utilized seismic wave interferometry to depict variations in the seismic velocity and consequently speculated on the pre-eruptive deformation

of a magmatic/hydrothermal reservoir (González et al., 2016). Matthews et al. (1997) proposed a continuous deepening of the crater floor associated with a high rate of degassing from fumaroles within the active crater; this degassing phenomenon was subsequently identified by an InSAR investigation (Pavez et al., 2006) and further confirmed by continuous monitoring using the high-resolution German satellite TerraSAR-X (Richter et al., 2018), which also showed that the deformation rate appeared to be largely unaffected by the most recent explosive eruptions in 2013 and 2015. A shallow hydrothermal system depicted by

magnetotelluric and seismic data (Díaz et al., 2012; Hellweg, 2000) seems to influence the active degassing at Lascar (Bredemeyer et al., 2018; Tassi et al., 2009). These degassing processes have been suggested to be the source of the tremors observed during the 1994-1995 period of unrest (Hellweg, 2000), and they are probably associated with the increase in long-period (LP) events preceding an eruption (González et al., 2016).

Since the end of 2010, a volcano monitoring network has been gradually deployed around Lascar, providing a reliable database

with which to investigate the 2015 eruption. In an attempt to better understand the most recent and unexpected eruption at Lascar, we compiled seismology, webcam, gas chemistry and thermal remote sensing data and conducted drone overflights after the eruption. By using this multiparametric data set, we study the details of the period preceding this eruption and describe the processes that might characterize other events at Lascar volcano. This work suggests a very shallow process culminating in steam-driven explosions after a heavy precipitation event. Our study also aspires to demonstrate why this type of volcanic

hazard is generally so challenging to forecast.

## 3 Data and analysis methods

A unique multidisciplinary data set comprising seismic data, optical images, gas compositions, and temperature and precipitation data, as well as unmanned aerial vehicle (UAV) images of the crater, is available to document the 2015 eruption. Lascar has been monitored by a network consisting of broadband seismometers, fixed cameras, permanent gas monitoring

equipment, and a local weather station, and this network has been further complemented by satellite-based infrared observations. The locations of all ground-based stations and the period analysed in this study are illustrated in Figure 1. The data treatment and analysis methods implemented in this study focused on the evolution of seismic volcano-tectonic (VT) and LP events, sulfur dioxide ($SO_2$) fluxes, $CO_2$ mixing ratios and temperatures of fumarolic emissions and changes in their brightness, thermal changes in the crater, and changes in the surface features of the crater. The instrumentation, data and

analysis methods are described in detail below.



### 3.1 Seismic monitoring

The network consists of five permanent seismic stations operated by the Observatorio Volcanológico de los Andes del Sur (OVDAS), the national volcano observatory that continuously monitors over forty active volcanoes in Chile. The OVDAS stations are equipped with broadband RefTek 151-30A sensors and RefTek 130B data loggers that record ground motions at a sampling rate of 100 Hz. Four of these five stations were used for the long-term seismic compilation employed in this study, and their locations are illustrated in Figure 1. However, only one seismic station was operational throughout the month of the eruption (station QUE; Figure 1).

To investigate the evolution of seismicity at Lascar, we used the entire available seismic catalogue from the volcano observatory. The catalogue was constructed by real-time scanning following Lahr et al. (1994) through visual classification performed by OVDAS analysts. The signal of each waveform was classified according to its spectral content, harmonic signature and time duration into one of six categories of signals observed at Lascar: VT, LP, VLP, volcanic tremor (TR), hybrid (HB) and explosion (EX) signals (Chouet, 1996). For our analysis, we considered VT and LP events since they were the most abundant events that occurred during the pre-eruptive phase from July 2014 to December 2015. Based on these data, we visualized the temporal evolution and characterized the precursor VT events as well as the eruption signal. This event analysis provided us with a fundamental overview of the stage of activity at Lascar volcano. For a more in-depth seismic analysis, the reader is referred to Gaete et al. (submitted manuscript, 2019).

### 3.2 Visual data

Solar-powered time-lapse camera stations were installed at two fixed locations to take pictures of the volcanic edifice at regular time intervals. Cameras C1 and C2 look to the northern flank of Lascar. C1 is a streaming webcam operated by OVDAS and is programmed with a capture interval of one image every minute (768 x 576 px resolution). Camera C2 is a high-resolution digital single-lens reflex (DSLR) camera (2,048 x 1,536 px resolution) from the German Research Centre for Geosciences (GFZ) that takes images at an interval of 60 minutes. Data from C1 and C2 (see Figure 1 for their location) were used to assess the height and duration of volcanic plumes using a kymograph in conjunction with pixel brightness analysis, as explained below.

Vision-based techniques have revealed the value of video data in a variety of applications, for example, in smoke and fire detection (Çetin et al., 2013; Healey et al., 1993; Verstockt et al., 2009). Camera-based monitoring networks have similarly become useful for observing plumes, lava movement and the ejection of particles during an eruptive episode (Brook and Moore, 1974; Chouet et al., 1974); in particular, the use of camera-based networks has increased dramatically in many fields of volcano research since their image quality and means of data transmission are continuously improving, thereby allowing nearly continuous monitoring, (Orr and Hoblitt, 2008; Paskievitch et al., 2006; Salzer et al., 2016; Walter, 2011). Water vapour is the major component of gases emitted from volcanoes, and the brightness of a volcanic plume has been shown to increase with the upwelling of new magma (Girona et al., 2015). To analyse the pixel brightness and variations in gas plumes, we





applied a correction using digital image correlation to ameliorate the shaking of the camera due to strong winds. Subsequently, we applied a masking operation to generate an image containing only the gas plume (see Supplementary material, Figure S1).

Each of these images was converted to greyscale and then transformed into a matrix, the elements of which represent the intensity values. We summed these matrix elements and normalized the values by the matrix size to compute the total brightness of one image. The results provided a measure of the pixel brightness variations occurring within the gas plume at Lascar volcano. To analyse two months of data, we utilized daily images generated at noon, which is when the visibility of the gas plume is the clearest and the position of the sun is constant. To perform a more detailed analysis, we used images produced

every hour. The pixel brightness can provide a useful visual constraint on the activity state of a volcano, even though the brightness alone may depend strongly on the position of the sun, condensation, and atmospheric conditions.

We further performed kymograph analysis by extracting the RGB values along a vertical line across the centre of the volcano and displaying these values in a time series plot with a data point every minute. The colouration, timing, duration and height of discrete gas emission pulses and eruption columns can be visualized with kymographs created in this fashion, and thus, they

have already been extensively used for the analysis of volcano video data in previous investigations (e.g., Munoz-Saez et al., 2015; Witt and Walter, 2017). A geometric conversion from the pixel scale to the metre scale was performed using the high-resolution Pleiades-1 tri-stereo digital elevation model presented in Richter et al. (2018). Due to the large distance from the cameras to the active volcano crater (6.75 and 6.42 km for C1 and C2, respectively), we applied a constant scaling factor to all pixels; thereafter, distortions arising from the lens, camera sensor, or field of view were not considered.

**3.3 Gas emissions**

Measurements of the $CO_2$ concentration and gas temperature were conducted at a low-temperature fumarole on the south-eastern crater rim of Lascar volcano (Figure 1). The gas temperature was monitored by means of thermocouples using an industrial platinum resistance temperature detector (RTD, PT 100, Labfacility, UK), and the $CO_2$ concentration was measured by means of a Vaisala CARBOCAP® Carbon Dioxide Probe (GMP343, Vaisala Oyj, Finland). The data from both sensors

were recorded at one-minute intervals with an ADL-MX Advanced Datalogger (Meier-NT GmbH, Germany). The analogue digital converter has an accuracy of 0.01%.

The $CO_2$ sensor, which has an accuracy of ±5 ppm, was successfully used in Chile in a previous study to correlate the volcanic activity with atmospheric changes (Zimmer et al., 2017). The temperature sensor, which can be applied in a range between -50 °C and +450 °C at an accuracy of ± 0.06 °C, was inserted to a depth of ~35 cm into the fumarole. Residual temperatures

were calculated by subtracting the mean temperature at the time of investigation from the actual reading.

Additionally, $SO_2$ emission rates were remotely monitored using a scanning mini-differential optical absorption spectroscopy (DOAS) station deployed by OVDAS 6.75 km north of the active crater (Figure 1); the mini-DOAS station yielded one complete scan across the gas plume every 5-15 minutes depending on the light conditions.

$SO_2$ slant column densities (SCDs) along the viewing direction of the mini-DOAS scanner were retrieved in the wavelength

range of 310-325 nm by means of DOAS (Galle et al., 2010; Platt and Stutz, 2008); the spectra measured inside the gas plume



were compared with a gas-free spectrum and the pixel-wavelength-calibrated $SO_2$ absorption spectrum from Vandaele et al. (1994), which was convolved with the slit function of the spectrometer. Furthermore, we incorporated an $O_3$ absorption spectrum (Voigt et al., 2001) and a Ring spectrum in the DOAS fit to avoid interference with absorption by ozone and scattering effects. Plume transport velocities and plume altitudes, which are required to calculate the gas flux from $SO_2$ SCD profiles,

were estimated using the wind speeds derived from archived weather data provided by the National Oceanic and Atmospheric Administration's (NOAA) Global Forecast System and using information obtained from pictures taken by the time-lapse cameras, respectively. The direction of plume transport was further determined by means of triangulation using the SCD profiles in combination with plume height estimates.

## 3.4 Thermal anomalies

To investigate thermal anomalies, we used satellite images acquired by Sentinel-2, an Earth observation mission from the European Union (EU) Copernicus Programme. Sentinel-2 was launched on June 23, 2015, though it became operational later in 2015; since then, it has systematically acquired imagery of Lascar volcano at a high spatial resolution (up to 10 m). The first available images were recorded in August 2015. Sentinel-2 acquires multispectral data comprising 13 bands in the visible, near-infrared, and short-wave infrared range of the electromagnetic spectrum. In this study, we utilized bands from the near-

infrared part of the spectrum to ascertain the variations in the appearance and dimensions of the lava dome inside the currently active crater of Lascar.

Here, we processed the aforementioned Sentinel-2 data using the SNAP toolbox (S2TBX), which is freely provided by the European Space Agency (ESA), and we performed a 12-11-8A band combination to derive a false-colour RGB image representing the apparent temperature of the ground. These bands allow the perimeter of the hot lava dome located inside the

Lascar crater to be mapped and further reveal the presence (or absence) of snow at the volcano summit. We then derived the approximate area of the dome through an analysis of the perimeter.

## 3.5 Aerial photography

UAVs were employed to obtain high-resolution nadir photographs from the bottom of the Lascar crater, which is hidden from the human eye by the rim of the crater. Due to the high altitude of the volcano (>5,500 m), the first successful drone flight,

that is, the first drone that did not crash, was launched in November 2017. The motivation of this overflight was to confirm the presence of a lava dome inside the active Lascar crater. We used a DJI Mavic Pro Platinum quadcopter drone equipped with a 12 MPx cropped-sensor camera (4,000 x 3,000 px, 35 mm focal length, ISO 100), which was programmed to take images at an interval of 2 seconds, and the flight speed was 5 m/s. The drone was launched from the southern rim of the active crater at 5,200 m above sea level (a.s.l.), after which it was elevated to 5,700 m a.s.l., and then it flew across the crater in a

northerly direction. From the 250 drone images acquired during the flight, we constructed a photomosaic and a hillshade using the structure from motion (SfM) workflow implemented in the Agisoft Metashape Professional software package. Over 20,000 tie points were identified for image matching, allowing us to generate a dense cloud consisting of over 7 million points; from



this point cloud, a digital elevation model with a 20 cm resolution and an orthomosaic map with a 7 cm resolution were produced. As no ground control could be obtained, we relied on the geolocations of the DJI geotagged camera images only, 230 leading to an image error estimation of 0.6-1.3 m and a total error of 1.25 m.

## 3.6 Weather data

Hydrometeorological conditions were monitored by a weather station (Vaisala WXT520) located at the base of the volcano (Figure 1); this station recorded the atmospheric pressure and temperature conditions, wind direction and wind speed, humidity, and intensity and accumulated amount of precipitation at a sampling rate of 1/minute. The data acquisition was temporally 235 synchronized by means of a Global Positioning System (GPS) device, and the data were collected in the field by a WiFi network and transmitted both to GFZ in Germany and to OVDAS in Chile.

We considered the intensity and accumulated amount of precipitation measured in a one-minute running average derived from samples acquired every 10 seconds with a range from 0 to 200 mm/h. The data were compared with the other observations to identify a rare precipitation event shortly before the 2015 explosion.

## 240 4 Results

First, we will describe the gradual changes leading up to the eruption; second, we will document the changes associated directly with the eruption; finally, we will discuss the data retrieved in the aftermath.

### 4.1 Gradual changes prior to the eruption

We analysed the seismic catalogue over the period from July 2014 to December 2015 to obtain an overview of the activity 245 preceding the eruption at Lascar (Figure 2a). In total, 1654 LP events and 47 VT events were identified during this observation period. A gradual increase in the number of LP events started in October 2014, i.e., approximately 1 year before the eruption. This increase in LP events was not associated with any relevant changes in the VT event rate. The peak of the LP event rate was found during April-July 2015, i.e., approximately 3-6 months before the eruption. In these months, the LP activity declined from ~11 events to ~1 event per day. At the same time, the persistent thermal anomaly on the crater floor observed in the 250 Sentinel-2 imagery gradually decreased in both size and intensity (Figure 3a,c), confirming an apparent decrease in activity. Precipitation is rare in this desert area with an annual average below 100 mm (Messerli et al., 1993). Nevertheless, from the continuous precipitation records recorded at the weather station situated at the base of the volcano, we observed three precipitation events occurring in the period analysed in this study (P1, P2 and P3, Figure 2a). Events P1 and P2 were characterized mainly by snowfall in the middle (March 2015) and end (August 2015), respectively, of the increasing LP activity 255 phase. In contrast, P3 occurred from October the 19th to the 21st of 2015 with 112 mm of accumulated rainfall, reaching 13 mm/h on October 20, 2015, only ten days before the eruption. This precipitation event was forecasted by the Chilean Meteorology Agency, prompting an emergency alert of yellow on October 18th from the Chilean National Emergency Office





(ONEMI) due to the severe weather warning throughout the entire region of Antofagasta; awaiting the maximum level of rainfall to be recorded over the Andes Mountains in the first 12 hours of October 19, 2015 (ONEMI, 2015).

Events P1 and P2 did not lead to detectable changes at the volcano. In contrast, event P3 entailed a pronounced increase in plume visibility, as was reflected by the strongly enhanced pixel brightness in the atmospheric column above the active crater. Thus, focusing our analysis on the daily changes of the plume to one month before and after the eruption on October 30, 2015, we observed low pixel brightness variations during the pre-precipitation phase (Figure 4a). Just after event P3, the pixel brightness suddenly increased, indicating the presence of abundant condensed water within the gas plume on October 21, 2015

(Figure 4a), and the maximum pre-eruptive peak was reached on October 23, 2015. These findings were confirmed upon closer inspection of the hourly variations in the brightness (Supplementary material, Figure S2), and the maximum value was revealed to be 4 times the normal state. Figure S2 also displays the daily cyclic fluctuations of the pixel brightness; these fluctuations were confirmed through visual analysis of the height and brightness of the plume in Figure 5d, in which a stronger signal is observed during the morning hours before transitioning to a slight manifestation of the plume in the afternoon.

Furthermore, the temperature of the fumarolic gas emissions clearly dropped from 33 °C to 29.2 °C, and some slight variations were evident in the $CO_2$ concentration after precipitation event P3; both of these trends seemed to be insignificant with respect to the amplitude variations in the long-term trends of their corresponding time series (Figure 4b,c). Similar to the fumarolic temperature, the $SO_2$ emission rates during October decreased from an average of 4.14 kg sec$^{-1}$ $SO_2$ in the period prior to event P3 to merely 2.33 kg sec$^{-1}$ $SO_2$ in the eight days following event P3 (Figure 4d). A renewed pulse of VT events began on

October 23rd and continued over the following days until the eruption (Figure 4e).

### 4.2 The phreatic eruption on October 30, 2015

The phreatic eruption that occurred on October 30, 2015, was first reported at 9:32 local time (12:32 UTC); the explosion expelled an ash plume that rose to approximately 1.7 km above the crater and then drifted towards the northeast, as revealed by the webcam and kymograph (Figure 5b).

Approximately 25 minutes before the explosive eruption, a cluster of 4 VT events was recorded within a 5-minute time window (Figure 2b). These VT events lasted for less than 10 seconds each, and the corresponding signals displayed characteristic frequencies in the range from 1 to 30 Hz. The eruption was characterized by the sudden onset of a harmonic tremor signal that lasted for approximately 50 minutes; the most energetic phase transpired during the first 20 minutes (Figure 2b,c). This signal was characterized by a frequency content between 1 and 10 Hz and a dominant frequency of 4 Hz; the eruption featured two

parts, the modulation of which seems to depict two phases (Figure 2c). The first part of the eruption signal lasted for 5 minutes and decreased strongly both in amplitude and in energy during the last 2 minutes. The second part lasted for approximately 10 minutes and was characterized by a modulation, i.e., a slowly decreasing amplitude (Figure 2b,c). Throughout the remaining 30 minutes of the eruption, the tremor exhibited a low energy and amplitude.

The phreatic eruption was accompanied by a degassing pulse, which was reflected by an 8-fold increase in the $SO_2$ emission

rate (Figure 4d, 5d). The average $SO_2$ emission rate was 6.76 kg sec$^{-1}$ during the morning hours prior to the eruption event,





and the first $SO_2$ emission peak of 55.14 kg sec$^{-1}$ was reached at 9:26 local time (12:26 UTC), which was approximately 6 minutes prior to the onset of the phreatic explosion. The corresponding optical $SO_2$ densities in the plume centre (i.e., the $SO_2$ concentrations along the viewing direction of the mini-DOAS scanner) started to increase considerably from 200 to 650 ppm*m approximately 15 minutes before the eruption and reached a maximum of 766 ppm*m approximately 5 minutes in advance of

the ash emissions (Supplementary material, Table S1). Thus, the 4 VT events that occurred between 25 and 20 minutes before the eruption were almost immediately followed by a vigorous increase in degassing activity. Subsequently, the $SO_2$ flux reached a second maximum of 44.38 kg sec$^{-1}$ at 10:05 local time (13:05 UTC), approximately 32 minutes after the onset of the eruption, and then gradually declined until it eventually returned to its pre-eruptive value of 6.7 kg sec$^{-1}$ at 10:59 local time (13:59 UTC), i.e., approximately 29 minutes after the eruption tremor ceased. The corresponding $SO_2$ SCDs similarly reached

a second maximum of 650 ppm*m concurrently with the fluxes before gradually diminishing to usual values. In contrast, the $CO_2$ concentration and temperature of the summit fumarole abruptly decreased from 4320 ppm $CO_2$ and 31.2 °C, respectively, in the evening hours of October 29th, reached local minima of 4250 ppm $CO_2$ and 29.2 °C throughout the duration of the eruption, and subsequently increased again to 4320 ppm $CO_2$ and 30.8 °C in the evening hours of October 30th (Figure 4b,c). The co-eruptive $CO_2$ minimum was further accompanied by a pixel brightness peak, exhibiting the presence of condensed

water in the eruptive plume (Supplementary material, Figure S2); this dynamic is a recurrent feature in the entire period illustrated in Figure S2.

A strong steam signal was observed in the two cameras beginning early in the morning of October 30th, as was also depicted by kymographic analysis (Figure 5d). A cyclic daily variation in the steam plume, which decayed in height and intensity from late morning to afternoon, was evident. During the explosion, the plume turned grey and exhibited three phases determined by

a change in the colour and the altitude. At the start of the eruption, there was a short lapse when the plume was light grey and reached ~6.3 km (P-I in Figure 5d). This lapse was followed by the main eruptive phase, during which a dark grey and taller plume (~7 km a.s.l.) evidencing that a high ash content was expelled (P-II in Figure 5d). These two pulses are associated with the two parts we observed in the eruptive tremor signal described above. The final phase exhibited an isolated light grey pulse that reached ~6.7 km (P-III in Figure 5d). This third pulse seems to stand alone, and no change in the seismic signal was

registered in this regard.

### 4.3 Post-eruptive observations

The seismic, thermal, and degassing anomalies detected shortly before and during the October 30, 2015, eruption very quickly returned to their background levels. For instance, the number of LP seismic events per day decreased to few or no events and displayed a very low peak-to-peak amplitude (Figure 2a, 4e). The Sentinel-2 data show a hot spot centred in the active crater

that revealed an increase in the size and intensity of the surface thermal anomaly on the crater floor in the first acquisition after the eruption (recorded on December 6, 2015; Figure 3d). This initial high temperature of the crater floor is also reflected by the camera images showing a glow at night after the eruption (Figure 5c). As the thermal anomaly zone at the crater floor gradually cooled (as observed in the Sentinel-2 data), the dimension and strength of the thermal anomaly slowly declined





during 2016 (Figure 3e-h). By the end of 2017, the thermal anomaly returned to the levels approaching those observed prior
to the 2015 eruption (Figure 3i,k).

The gas data displayed pronounced changes at the fumarole monitored on the crater rim. We first found an increase in the
temperature from 29.2 °C to 34.2 °C (in the 3 days after the eruption until November 2, 2015) and an increase in the $CO_2$
concentration from 4250 to 4370 ppm (until November 5, 2015), which gradually approached their background levels again
~5 days after the eruption (Figure 4b,c). According to an analysis of the camera brightness, the degassing also gradually
returned to its background levels by November 9, 2015, when they dropped to usual pre-precipitation values (Figure 4a). The
$SO_2$ fluxes during the period of November 2-5, 2015, were measured at an average of 4.42 kg sec$^{-1}$ and therefore were slightly
enhanced with respect to the average (2.32 kg sec$^{-1}$) during the entire post-eruptive period considered here (October 31, 2015,
to November 31, 2015) (Figure 4d).

Our UAV overflight performed on November 27, 2017, revealed the presence of a dome-shaped elevation located at the base
of the deep crater floor with a diameter of ~57 m and a blocky surface representing the remnants of the 2015 lava dome (Figure
7a,c). The side flanks and talus apron of the lava dome were partly covered by rock fall deposits from the Lascar crater walls.
We compared the location of the lava dome to a thermal anomaly map acquired during the 2015 eruption, and good agreement
was observed between the blocky crater floor and the thermal anomaly region (Figure 7a,b). Close-up views enabled by high-
resolution drone photogrammetry further revealed the presence of a linear feature striking NE-SW dissecting this lava dome
but not dissecting the apparently younger rock fall deposits (Figure 7d). The dissection of lava domes by linear features has
commonly been observed elsewhere following steam-driven explosions (Darmawan et al., 2018a; Walter et al., 2015).
Therefore, we speculate that a linear NE-SW-striking fracture developed during the 2015 steam-driven explosion.

**5 Discussion**

The steam-driven explosive eruption of Lascar on October 30, 2015, was the first that was densely monitored. The eruption
was studied by utilizing different data streams, the results of which suggest that (i) no magma movements within a shallow
magma reservoir were identifiable immediately prior to the explosion and (ii) the spontaneous steam-driven explosion was
directly associated with short-term degassing and the development of a fractured dome. We ascertained that the volcano was
in an elevated stage of activity, as the steam explosion was preceded by ~1 year of LP seismic activity. However, as the seismic
activity gradually declined approximately 4 months prior to the explosion (Figure 2a), a direct and causal relationship is
debatable. We noticed that this decline in seismic activity was accompanied by a reduction in the persistent high-temperature
anomaly located inside the active crater, which was associated either with fumarole activity or with the extrusion of magma
(Figure 3a-c). Similar decreases in the area and intensity of hot spots have previously been observed preceding, e.g., the
eruptions of February 1990, December 1993, and July 1994 (Oppenheimer et al., 1993; Wooster and Rothery, 1997). The
details of our findings, limitations and interpretations as well as a conceptual model will be discussed in the following.



## 5.1 Water infiltration into a cooling lava dome at Lascar

Different processes may drive phreatic eruptions; for example, magma may intrude wet sediments and aquifers, lava or pyroclastic flows may interact with surface water, or hydrothermal systems may form during periods of repose (Barberi et al., 1992; Rouwet et al., 2014). Moreover, evidence that rainfall can trigger volcanic activity has also been documented, such as the dome collapse at Soufrière Hills Volcano, Montserrat (Carn et al., 2004; Matthews et al., 2002), the seasonal response of the seismic velocity (Sens-Schönfelder and Wegler, 2006), the increase in seismicity associated with degassing process influenced by the precipitation at Merapi volcano (Richter et al., 2004), and the possible phreatic eruptions that induce dome collapse (Darmawan et al., 2018a). Rainwater or meteorological forcing has hitherto not been identified at Lascar, and our data also leave room for speculation and questions. Nevertheless, we note that the historic activity of Lascar provide evidence for at least 6 phreatic eruptions and that approximately 50% of all recorded eruptions occurred in the period from mid-September to mid-December following possible rainfall events (Table 1), thereby showing a possible seasonal dependence similar to that observed elsewhere in the Andes (Bredemeyer and Hansteen, 2014; Mason et al., 2004). Additionally, the October 2015 eruption falls within this period and occurred only a few days after a rainfall episode, which possibly led to the observed eruption.

Furthermore, we note that in our observation period, another two precipitation events occurred in March 2015 and August 2015 (P1 and P2, respectively, Figure 2); event P1 was larger than any other precipitation event that transpired during that year and occurred during a period of high (or even peak) seismic activity at Lascar. Why this event with such considerable precipitation did not trigger a phreatic explosion and why it was also notably not associated with any other changes in degassing or VT seismicity remain puzzling (OVDAS, 2015). Considering earlier studies on the explosive activity of lava domes, one might instead expect explosion(s) at the onset of lava dome extrusion thought to be promoted through a reduction in porosity caused by vesicle flattening and mineral precipitation (Boudon et al., 2015). However, our study showed that the explosive eruption at Lascar occurred during a phase of crater floor subsidence that was identified by InSAR (Richter et al., 2018); thus, a possible explanation may lie in the timing of event P3.

Event P3 accumulated 112 mm of precipitation in three days, which is still comparable to the average precipitation of 100 mm per year, as is typically observed at the Atacama Large Millimeter/submillimeter Array (ALMA) observatory (over 5000 m a.s.l. and 40 km north of Lascar volcano). This rainfall could likely percolate into a dome carapace that was already being subjected to cooling, possibly forming a deep fracture during the contraction of the lava dome body. For comparison, the most explosive eruptions elsewhere did not necessarily occur during peak volcanic activity and maximum precipitation but occurred merely during a period when the deep percolation of water was no longer inhibited by a growing lava dome (Darmawan et al., 2018b). In this view, at Lascar volcano, the deep percolation of water and the initiation of the phreatic explosion occurred not during the climax of volcanic unrest but when the lava dome was cooling and contracting.

Our analysis of different data streams over the one month prior to the eruption and the one month after the eruption allowed us to recognize precursory anomalies in the VT seismicity and the brightness of the volcanic plume; these anomalies led us to





suggest a relationship with the precipitation of event P3. The maximum pre-eruptive brightness was followed by the reactivation of VT activity, which was delayed one day, and was accompanied by continuous fumarole cooling in response to

the infiltration of external water; during this period, the cooling rate and depth of penetration were correlated with the amount of precipitation and soil cracks (Zimmer et al., 2017).

Phreatic volcano explosions typically occur without any precursors (Stix and de Moor, 2018). These types of eruptions are similarly believed to occur without precursors at Lascar, a volcano with frequent and well-documented phreatic explosions. Two different endmembers of phreatic eruptions have been identified: those associated with a deeper hydrothermal system

(type 1) and those associated with a near-surface hydrothermal system (type 2) that also includes surface waters. Lascar volcano is known for its lively history of violent hydrothermal explosions, and our observations suggest that the 2015 eruption was a type 2 phreatic explosion according to the Stix and de Moor (2018) classification scheme. Thus, we discuss a possible trigger for this event due to the infiltration of rainwater through cracks produced by a cooling carapace. This liquid water was stored at shallow levels and was vaporized by hot rocks, which led to the explosion.

A short-term increase in VT activity was observed in direct association with the eruption, after which the VT activity rapidly declined. The absence of VT events from the record almost two months prior to the eruption suggests that the event that occurred on the 24th and those that were maintained throughout the following days were signs of fracturing caused by an increase in the pressure that followed the percolation of rainwater into Lascar's hydrothermal system. For example, VT events preceded the 2007 phreatic eruption of Mount Ontake, Japan (Kato et al., 2015), which was associated with the infiltration of

hot fluids from the hydrothermal system. At Lascar, the short-term increase in VT activity occurred during the timing of an increase in the pixel brightness observed by the cameras; both of these phenomena may indicate the formation of new cracks exposing the previously sealed hydrothermal system.

Moreover, the addition of percolating water into the system could have led to a strong dilution or reduction in $CO_2$ emissions in response to an increased groundwater content exhibiting an anti-correlated relationship between the steam brightness and

$CO_2$ concentration (Supplementary material, Figure S2). This dynamic was previously discussed as being the result of an atmospheric pressure reduction governing the boiling temperature of water and consequently increasing the vaporization of water in shallow aquifers (Zimmer et al., 2017). The only comparison in this regard was performed for $SO_2$ (Girona et al., 2015), where a very similar trend was observed in the steam brightness and the $SO_2$ column abundance curves acquired simultaneously during a very short (30 minute) passive degassing period at Mount Erebus, Antarctica. The correlation between

the $SO_2$ and brightness time series reflects the fact that volcanic water vapour is typically emitted in proportional amounts to the $SO_2$ and $CO_2$ abundances in the plume in the short term and under normal conditions. This concept can provide an idea about the parametric relationship between the water and gas contents in fumaroles. For our results, the most plausible interpretation advocates that the percolating water added into the system needed some time to heat up. Therefore, we observe a gradual increase of vaporized water over the five days preceding the eruption, the excess of which is evidenced in the

relatively white signature of the fumarole steam. The gradual increase observed during the three days immediately following the rainfall likely can be associated to vaporization of water falling in the near field of the dome (Figure S2).





The rapid fumarole heating after the eruption and the short-lived increment in the $CO_2$ concentration (Figure 4b,c) may be interpreted as a response to the open conduit resulting from the fracture truncating the dome due to the eruption. Previous studies have described rainfall-triggered structural destabilization and lava dome collapse and their thermal-hydraulic

mechanisms (Elsworth et al., 2004; Hicks et al., 2010; Matthews and Barclay, 2004). The remaining linear feature observed across the lava dome that we believe formed during the 2015 eruption could provide evidence of a pressure build-up and an explosion of the system (Figure 7b). This would allow for the existence of a persistent thermal anomaly, as is evidenced from satellite data even after one year (Figure 3) and the appearance of a glow from the crater the night after the eruption (Figure 5), which was still observable on the night of November 2nd according to the kymographic analysis (Supplementary material,

Figure S3a). This also favours the mechanism of degassing transporting more energy through hot fumarolic gases as a result of a higher magmatic gas flux into the atmosphere. The entire gas pulse accompanying the eruption expelled at least 170 metric tons of $SO_2$ in only 100 minutes, which is approximately half the amount of $SO_2$ that Lascar volcano usually emits throughout a whole day in a non-eruptive period (Tamburello et al., 2014; Bredemeyer et al., 2018). Using the molar ratios of Tamburello et al. (2014), we additionally obtain equivalents of at least 5,440 metric tons of $H_2O$ and 230 metric tons of $CO_2$ that were

released during these 100 minutes.

This large mass of emitted gas detected from the ground was nevertheless dwarfed by the estimated 49 kilotons of emitted $SO_2$ recorded from the eruption cloud by the Ozone Monitoring Instrument (Global Volcanism Program, 2016). The enormous discrepancy between these two measurements may be attributed to the high aerosol contents (condensed water droplets and ash) that were visible particularly in the proximal portion of the volcanic plume but rapidly diminished with increasing distance

from the emission source (Figure 5b) due to the downwind evaporation of water droplets and the gravitational settling of ash. It is thus very likely that the near-field ground-based measurements recorded at a distance of approximately 5 km from the source were significantly more affected by light scattering on the surface of the volcanic cloud than were the far-field satellite observations, leading to a severe underestimation of the light absorbance caused by $SO_2$ in the cloud (Mori et al., 2006). This so-called light dilution effect was also reflected in the fact that the $SO_2$ column densities in the centre of the volcanic plume

(Supplementary material, Table S1) were lower during the ash emission period than they were prior to the eruption.

**5.2 Limitations of the used methods**

Locating seismic events with a reliable accuracy largely depends on how dense the seismic network is and its spatial distribution. Throughout October 2015 and thus during the eruption, only one seismic station was operational (QUE, see Figure 1). Hence, a limitation during the processing/analysis of the seismic data was that we could neither constrain the locations of

the VT and tremor signals nor characterize the event sources due to a lack of data; moreover, because the events were characterized by a small magnitude, they were strongly affected by noise.

Satellite remote sensing has continuously developed in recent decades to include better sensor technology and higher-accuracy measurements, thereby showing that it is a very useful technique, especially for monitoring remote areas that are difficult to access. Nevertheless, the dependence on the weather conditions to obtain good-quality images and useful information from





satellite images still remains one of the main limitations of this approach, which we have also mentioned as an issue in our observations.

The main limitation of the methods used for computing the brightness and performing kymographic analysis is that they require good atmospheric conditions and a wind direction that favours the visibility of the plume within the image field of interest. In particular, pixel brightness analysis requires a permanently visible volcanic plume (preferentially under blue sky conditions)

to acquire clear and good-quality images. Nevertheless, although our method does not consider the influences of long-term atmospheric variables (e.g., pressure, humidity and temperature) governing the condensation of water vapour in a volcanic plume, the resulting perturbations can be corrected as long as they are well defined. The Sun's position and diurnal radiation variations additionally cause cyclic perturbations, which can be either avoided by comparing pictures taken at the same time of day or corrected by analysing reference regions, as proposed by Girona et al. (2015). The use of this approach implies an

advantage with respect to other established gas monitoring methods considering its low cost and its straightforward and valuable implementation for detecting changes in a volcanic plume.

$SO_2$ flux measurements from the mini-DOAS scanner could be conducted only during daylight, when the UV intensity is sufficiently high, and therefore, as they contain gaps at nighttime, these records are semi-continuous. The measurements at Lascar were further restricted by the fact that only one scanning mini-DOAS station was installed 6.75 km away, with a target

on the northern face of the crater, thus limiting the observation range to a northerly direction of plume transport that was confined between WNW and ENE. Thus, we were fortunate insomuch that the eruption plume drifted through the field of view of that one station, enabling us to conduct this detailed study of its optical properties. Volcanic plumes that drifted in directions outside this range were not captured by the instrument, causing additional discontinuities in the gas emission time series. Moreover, such measurements are strongly affected if considerable amounts of ash are present in the volcanic plume, leading

to a severe underestimation of the emitted gas mass, as discussed above. Likewise, the passage of meteorological clouds during the rainfall period likely introduced larger uncertainty into our observations due to the scattering of light on the water droplets in the cloud, leading to either an underestimation or an overestimation depending on the locations of those meteorological clouds with respect to the volcanic plume (Kern et al., 2010). Stormy conditions, which were frequently encountered throughout the entire observation period, caused a strong dilution of the gas plume already located in the near-field region of

the volcano, which resulted in a reduction in the already very low $SO_2$ SCDs, often hindering the detection of the gas plume.

Good and accurate in situ gas and temperature measurements depend on the appropriate placement of the instruments as a consequence of the accessibility of those locations. To this end, measurements of the $CO_2$ variability at Lascar could be carried out only at the fumarole field on the rim of the crater, where magmatic gas emissions are altered upon mixing with hydrothermal fluids. Moreover, observations from the crater rim may not be representative of the gas discharge conditions elsewhere

throughout the volcano, particularly when the focus of gas emission changes. The gas emissions during the eruption, for example, were strongly focused on the crater area, suggesting that the marginal areas at higher elevations (e.g., our fumarole on the crater rim) were less well supplied with gas because the overpressure was released elsewhere. The co-eruptive $CO_2$ minimum was thus likely a consequence of the location of the $CO_2$ sensor, which is one major limitation of using point-





sampling devices. Furthermore, the $CO_2$ mixing ratio and temperature of a fumarole are also strongly susceptible to changes
in the weather, which is why the interpretation of these variables is often complicated when observed trends cannot explicitly
be attributed to weather conditions or volcanic activity. Monitored together with relevant weather variables, however, the $CO_2$
mixing ratio and temperature are extremely well suited for studying the influence of weather conditions on volcanic activity.
Finally, using drones provides the possibility of attaching equipment to measure different parameters (e.g., thermal anomalies,
gas emissions, and structural features) from an overflight. However, the successful use of drones is usually limited by
complicated access to the remote areas where volcanoes are located. Specifically, the Lascar's summit is located at an altitude
of almost 6 km a.s.l., and thus, climbing to this requires a preceding acclimatization phase due to the thin air conditions.
Furthermore, overflights are usually affected by strong winds, which are very common at such high altitudes, implying that a
drone will crash or directly preventing a drone from being launched. This was the main reason for the 2-year delay in our data,
which is why some changes in the crater floor associated to the eruption may have remained undetected.

## 5.3 Implication for future monitoring instrumentation

Seismic activity often increases before an eruption; hence, its monitoring is a good method for forecasting the risk of a volcanic
eruption (Tokarev, 1963). Nevertheless, predicting spontaneous explosive activity remains a challenge either because dense
instrumentation is available or because certain eruption types, particularly steam-driven explosions, are not preceded by
seismicity, although a few examples suggest that precursors may also exist for this eruption type. A good example is Aso
volcano, where a steam-driven explosion was preceded a few minutes earlier by a LP displacement signal due to the inflation
and deflation of the source, which could provide warning for such eruptions detected by broadband seismometers (Kawakatsu
et al., 2000). Nevertheless, many manifestations of activity are easily detectable considering variables beyond seismic activity.
In the case of Mount Erebus, a multiparametric monitoring network was deployed, and it was found that strombolian explosions
are accompanied by acoustic and irradiance pulses and seismic VLP signals (Aster et al., 2004). The use of combined data sets
enables the complex processes of volcanoes to be approached and allows their activity, even in remote locales where the
hostility of weather makes such monitoring a challenging task, to be better understood.

From this perspective, the apparent absence of short-term precursors from the historical activity reported at Lascar could be a
consequence of the network deployment. Comparing the observations derived from the complete image data set covering the
2015 eruption, we were able to find evidence of long-term LP variations starting with an 8-month increase followed by a 4-
515  month decrease that was accompanied by a constant drop in the thermal anomaly observed in August 2015. Likewise, shorter-
term changes were also observed in the steam, followed by the occurrence of new VT events, the frequency rate of which
increased only a few minutes before the explosion. Overall, these observations allow us to infer a preparatory phase preceding
the eruption. We speculate that to maintain continuous degassing and sustain a persistent anomaly at Lascar, it is necessary to
maintain minor magmatic feeding from greater depths. Nevertheless, this feeding may have been affected by a partially blocked
520  fluid path, as can be concluded from the data acquired before the 2015 steam-driven explosion, which in turn could have
destabilized the shallow hydrothermal system, eventually inducing surface activity that declined after the eruption. The closure



of the hydrothermal system can be explained by the decrease in the thermal anomaly and LP activity, which could be associated with crater subsidence (González et al., 2015; Richter et al., 2018).

Finally, most of the methods exposed in this study are broadly and successfully used to monitor volcanoes; for instance, drones are employed to map morphological structures (e.g., Darmawan et al., 2018a), mini-DOAS scanners are used to measure $SO_2$ emission rates (e.g., Bredemeyer et al., 2018), satellite data are utilized to detect thermal anomalies (e.g., González et al., 2015), and in situ sensors are used to record $CO_2$ concentrations and fumarole temperatures (e.g., Zimmer et al., 2017). In addition, the use of optical images from permanently installed cameras in this study was shown to be an alternative way to detect pre-eruptive changes through pixel brightness analysis and to estimate the variations in and the altitude of the eruptive plume using kymographic analysis, which will be very useful for forecasting steam-driven eruptions at Lascar and other volcanoes taking into account the limitations of this approach. Furthermore, the use of drones equipped with different sensors, such as high-resolution optical systems, allowed detailed information about changes that otherwise would have remained concealed to be revealed.

**5.4 Conceptual model of the 2015 Lascar phreatic explosion**

Our interpretation inferred from the parametric relationship between the $CO_2$ concentration and residual temperature of the fumarole after the percolation of meteoric water into the lava dome and its underlying hydrothermal system is illustrated in Figure 6. Here, we speculate about the cyclic evolution of the $CO_2$ mixing ratio and residual temperature of the summit fumarole two weeks after precipitation event P3. The cycle started with a reduction in the residual temperature of 4 °C during the period of October 19-30, 2015, and minor variations in the $CO_2$ content. The following three days after the eruption, the $CO_2$ mixing ratio changed from 4240 to 4380 ppmv, which was also associated with a moderate increase in the residual temperature. During November 4-7, 2015, the residual temperature further increased, whereas the $CO_2$ concentration decreased and slowly returned to the "normal" pre-precipitation values (~4260 ppmv $CO_2$ and 0 °C residual temperature). Figure 6 clearly shows the effects of meteoric water on the evolution of the $CO_2$ concentration and the fluctuations in the fumarole temperature. In addition, the 3-day delay in the increase in the steam brightness seems to be the required time for water to infiltrate into the hydrothermal system and to significantly change the composition of the visible fumarole exhalation.

The apparent anti-correlated behaviour of the steam brightness and $CO_2$ concentration (Supplementary material, Figure S2) may suggest that an increase in the water vapour within the plume is directly associated with a lower $CO_2$ concentration at fumaroles. However, similarly as $SO_2$ and $CO_2$ concentration measurements from fumaroles are known to be influenced by air pressure and wind speed changes (e.g., Nachshon et al., 2012; Zimmer et al., 2017), the brightness changes we observed in the images recorded by our cameras strongly depend on the relative humidity, air temperature, pressure and wind speed (Girona et al., 2015) favouring condensation of water droplets and thus the visibility of the volcanic cloud during the early morning hours. Thus the daily trends of observed brightness peaks are likely rather caused by changing ambient conditions, while the long-term trend of this proxy reflects a measure of the amount of water vapour emission. This is further corroborated by the fact that the $SO_2$ emission rate did not show any pattern associated with changes in the fumarole brightness, although the $SO_2$





emission rate is usually much more affected by scrubbing through hydrothermal water than is $CO_2$ (Symonds et al., 2003). The observed difference between the observed behaviour of $SO_2$ flux and $CO_2$ concentrations however, also can be associated with distinct instrument observation ranges. While $SO_2$ fluxes were measured globally from the entire volcano and were largely dominated by gas contributions from the plume emitted mainly from the magmatic dome, the $CO_2$ concentration was measured locally at a fumarole located along the crater periphery. The scrubbing of $SO_2$ emissions is thus less likely to occur at Lascar

since the gases emitted by the dome do not have to pass through the aquifers of the surrounding hydrothermal system (Bredemeyer et al., 2018), whereas the $CO_2$ concentrations on the crater rim may indeed have been affected by water infiltration of the volcanic edifice. In the long term, anyway, both parameters (i.e., the $SO_2$ and $CO_2$ concentrations) did not display significant variations that deviated from the normal previously observed behaviour.

The previous interpretation and the processes preceding the eruption are conceptualized in Figure 8. Our conceptual model

shows a pre-eruptive phase (Figure 8a) characterized by a sustained increase in LP seismic activity. We interpret this phase as the sign of a sustained fluid injection from depth inducing a long-lasting gradual pressure build-up due to a progressively blocked fluid path induced by a cooling lava dome. This interpretation is derived from the decrement in LP events and the reduction in the thermal anomaly evidenced in the images acquired by Sentinel-2 months prior to the eruption. The co-eruptive phase spanning 20 days started with precipitation event P3 on October 19-20, which caused the subsequent percolation of

water into the volcanic system (Figure 8b-1), which in turn reduced the temperature of the fumarole and eventually prevented efficient outgassing, as is depicted by the slight reduction in the $CO_2$ concentration (Figure 8b-2). This water was vaporized and released into the atmosphere, increasing the water content of the volcanic plume. The eruption (Figure 8b-3) principally released water, $CO_2$ and $SO_2$, thereby expelling much of the excess water within the hydrothermal system. On the following days, an increase in the temperature contributed to a further reduction in the water content in the plume, and the gradual drying

of degassing pathways led to the observed increase in the $CO_2$ concentration (Figure 8b-4). The post-eruptive phase (Figure 8c) is distinguished by a permanent thermal anomaly that slowly decreased in temperature throughout the following year.

## 6 Conclusion

This paper covered the period of unrest corresponding to the eruption of Lascar volcano on October 30, 2015, and represents the first attempt to characterize a steam-driven explosion of Lascar and its precursor activity despite the fact that such eruptions

occur therein on a regular basis. From a multidisciplinary data set, we found that signs of unrest and precursor activity can manifest as spatiotemporal changes on a variety of scales among the measured parameters. In the presented case, long-term changes in LP seismic activity preceded the eruption, but this was not observed afterwards, while gas and photography data displayed short-term variations in a time window ranging from some days prior to several days after the eruption. In addition, long-term remote sensing information allowed us to recognize a thermal anomaly after the eruption that helped to support our

short-term observations.



Overall, the multiparametric analysis suggests that the period of unrest began with a 1-year rapid increase in LP seismicity, which increased the pressure already built up within the volcanic system towards the climax stage, which was depicted by a drop in LP activity. During this time, an unusually strong precipitation event occurred that could have been the trigger of the eruption by acting on the dynamics of the system and inducing short-term parameter changes that evolved in 4 stages over a

time window of days as follows: 1) precipitation, 2) enhanced steam formation and VT activation, 3) eruption and 4) a temporary increase in the $CO_2$ mixing ratio and fumarole temperature. Afterwards, the thermal anomaly in the crater slowly decreased in intensity approximately 1 year after the eruption, thereby concluding the process associated with the 2015 eruption (see Figure 8).

Our analysis has shown that by combining different techniques and data streams, we can deploy an efficient monitoring system

to perform a successful hazard assessment and establish an early warning system for volcanoes that seem to show no clear precursor activity. In addition, we have demonstrated that signs of unrest and precursor activity can manifest as spatiotemporal changes on a variety of scales that should be promoted for future monitoring at Lascar and other volcanoes worldwide exhibiting similar behaviours.

### Acknowledgements

This research was supported by the German Research Centre for Geosciences, GFZ and by the program Forschungsstipendium für Doktorat from Deutscher Akademischer Austausch Dienst DAAD. This is a contribution to VOLCAPSE, a research project funded by the European Research Council under the European Union's H2020 Programme / ERC consolidator grant n. [ERC-CoG 646858]. We thank Stefan Mikulla, Jacqueline Salzer, Mehdi Nikkhoo, Elske de Zeeuw-van Dalfsen and Nicole Richter for the help, contributing to earlier field activities and preparation. We also thank our colleagues at OVDAS in Temuco and

Gabriel Gonzalez at UCN Antofagasta for discussions, and the analysts and technicians of the observatory for their laborious efforts to provide a complete seismic event database. A.G. and S.B. moreover appreciate and thus would like to express their gratefulness for the hospitality and support they experienced throughout their extended stays at the observatory. We additionally thank to ESA for providing Sentinel-2 data.



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





**Tables and figures**

**Table 1: Historical record of activity at Lascar volcano (Global Volcanism Program, 2013; González-Ferrán, 1995; Siebert et al., 2010)**


| Start | Duration | Type extrusion | Explosive | VEI |
|-------|----------|----------------|-----------|-----|
| 1902 | ---- | ---- | Y | 2? |
| 1933-Oct-09 | 3 Months | ---- | Y | 2 |
| 1940 | ---- | ---- | Y | 2 |
| 1951-Nov-16 | 3 Months | ---- | Y | 2? |
| 1954-Jun-16 | 1 Month | ---- | N | 2? |
| 1959-Nov-16 | 7 years, 2 Months | ---- | Y - Phreatic | 2 |
| 1969-May-16 | ---- | ---- | N | 1 |
| 1972-Jul-02 | ---- | ---- | ? | 2? |
| 1974-Jul-16 | 2 Months | ---- | ? | 1 |
| 1984-Dec-16 | 7 Months | dome building | N | 0 |
| 1986-Sep-14 | 2 Days | 15 km ash column | Y | 3 |
| 1987-Nov-16 | 2 Years, 5 Months | dome building and ash cloud | Y - Phreatic | 3 |
| 1990-Nov-24 | 1Day | 1.5 km  eruption columns | Y | 1 |
| 1991-Oct-21 | 7 Months? | dome collapse, dark plume and ashfall | Y | 2 |
| 1993-Jan-30 | 7 Months | dome collapse, plinian column, bombs ejection and pyroclastic flow | Y - Phreatic strombolian explosions | 4 |
| 1993-Dec-17 | 2 Months | dark grey plume | Y - Phreatic | 2 |





| 1994-Jul-20 | 6 Days | dark and brown columns and small ashfall | Y- vulcanian activity | 2 |
| 1994-Nov-13 | 8 Months | vapour and ash plumes, black plume | Y - vulcanian and small phreatic eruption | 2 |
| 1996-Oct-18 | 1 Day | white-bluish vapour emission | Y | 2 |
| 2000-Jul-20 | 6 Months | Ash emission | Y | 2 |
| 2001-May-17 | 2 Months | ---- | N | - |
| 2002-Oct-26 | 2 Days | 1.5 km ash plume | Y | 2 |
| 2003-Dec-09 | 1 Day | fine ash from fumarole | ? | 1 |
| 2005-May-04 | 1 Day | 10 km ash cloud | Y-vulcanian eruption | 3? |
| 2006-Apr-18 | 1 Year, 3 Months | white-grey plumes and ash emission | Y - Phreatic | 3 |
| 2013-Apr-02 | 7 Months | seismicity, glow and grey plume | Y | 1 |
| 2015-Oct-30 | 21 Days | 2.5 km ash plume variable seismicity | Y | 2 |
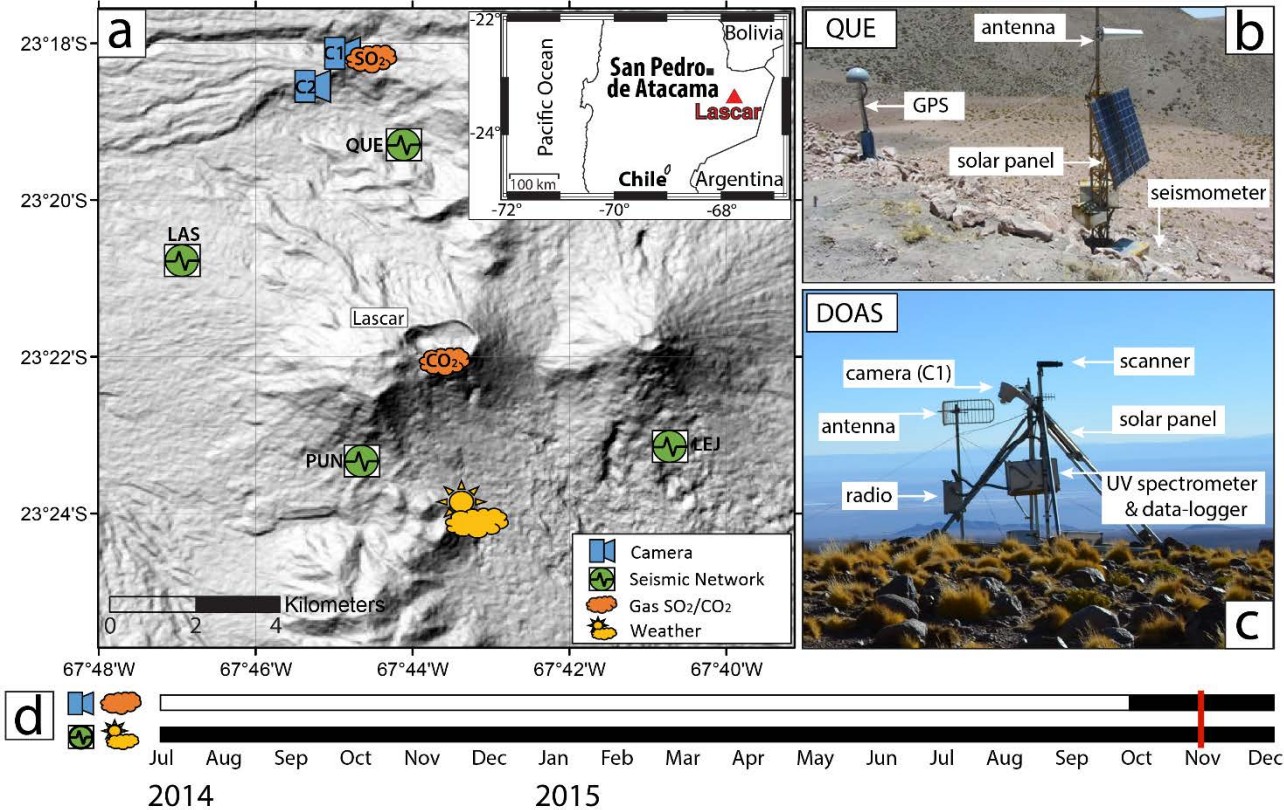

**Figure 1: a) Locations of Lascar and the deployed equipment applied in this parametric study. b) Setup of seismic station QUE. c)**
**Setup of the internet protocol (IP) camera (C1) and the mini-DOAS station to measure SO₂ emissions. d) Time bars depicting the**
**period analysed in this study using each data set. The red line indicates the eruption on October 30th.**



**Figure 2: a) Temporal evolution of seismicity and precipitation events from July 2014 to December 2015. Cumulative LP events**
**(*purple dots*) show a permanent increase starting one year before the eruption and a slight decrease three months before the eruption.**
**VT events (*green stars*) do not show any evolution pattern but cluster around the eruption. P1, P2 and P3 represent three**
**exceptionally strong precipitation events that occurred during the study period; P3 is the suggested trigger of the eruption. b) Seismic**
**record from October 30, 2015. The red box indicates the window covered in (c) and shows the VT (*i, ii, iii, v green arrows*) and LP**
**events (*iv purple arrow*) preceding the eruption and the eruption signal (*vi blue arrow*). c) Spectrogram depicting the frequency band**
**covered by the VT and LP events as well as the eruption tremor. The events that occurred early in the morning outside the red box**
**in (b) are regional tectonic events located more than 100 km away from the volcanic area; therefore, there is no certainty that they**
**could have affected the volcanic system.**
**Figure 3:** Satellite thermal anomalies (high temperatures are shown by red colours, while cold temperatures and snow are shown in blue) retrieved from Sentinel-2. The availability of data is irregular in 2015, but the data show a gradual decrease in the thermal anomaly in August 2015 and a later increase in the thermal anomaly on December 6, 2015, followed by a gradual decrease. Note that between August 28, 2015, and December 6, 2015, no clear Sentinel-2 images were acquired. Image (g) shows the thermal anomaly during the time of our UAV field campaign shown in Figure 7.



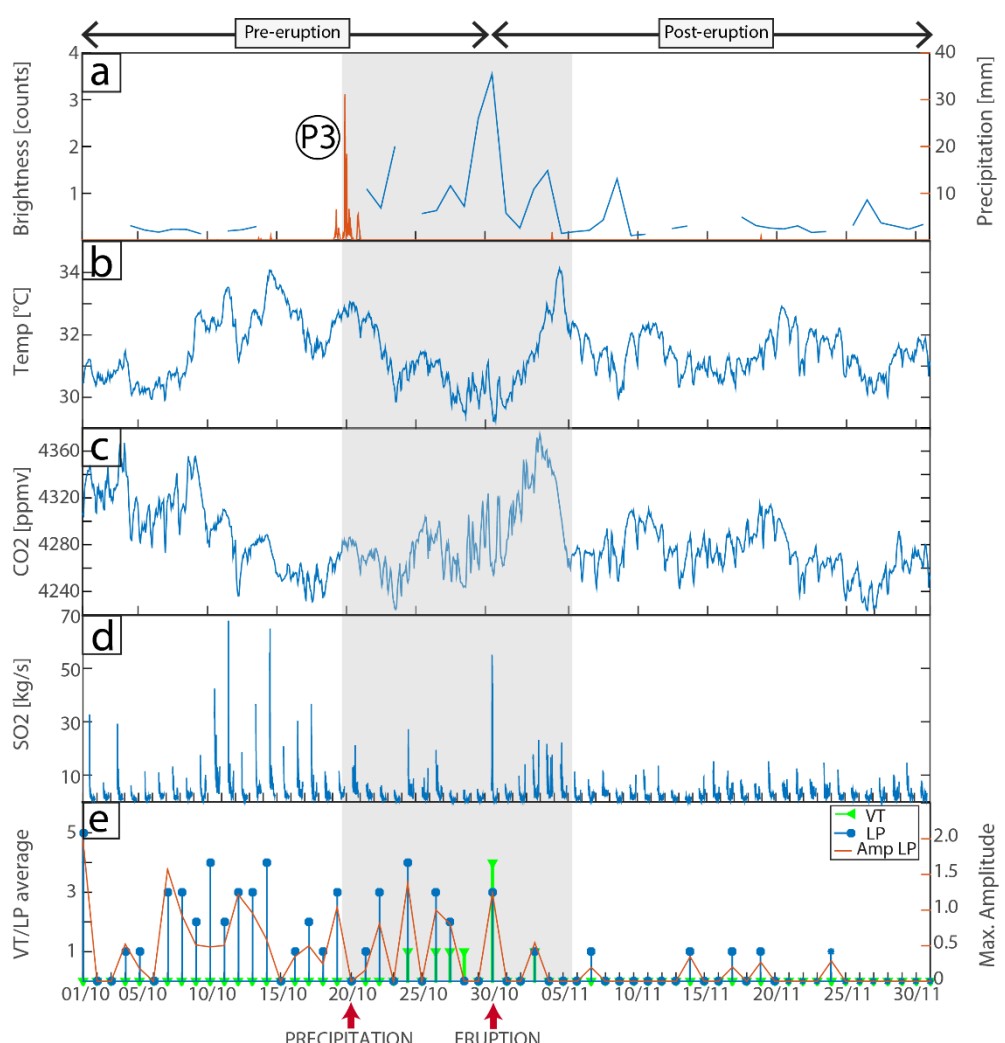

**Figure 4: Two months of daily variations shown in multiparametric plots of data. a) Brightness (*blue curve*) of the fumarole computed at noon and precipitation (*orange curve for P3, see Figure 2*), b) fumarole (*blue*) temperature, c) $CO_2$ mixing ratio, d) $SO_2$ emission rate (note that the data show a peak during the eruption), and e) VT/LP seismic event average and LP peak-to-peak amplitude (*orange curve*). Precipitation event and eruption days are indicated with arrows at the bottom. The shaded box indicates the period covered from the onset of precipitation event P3 to a few days after the eruption, during which high variability was observed in the gas plume and seismicity.**


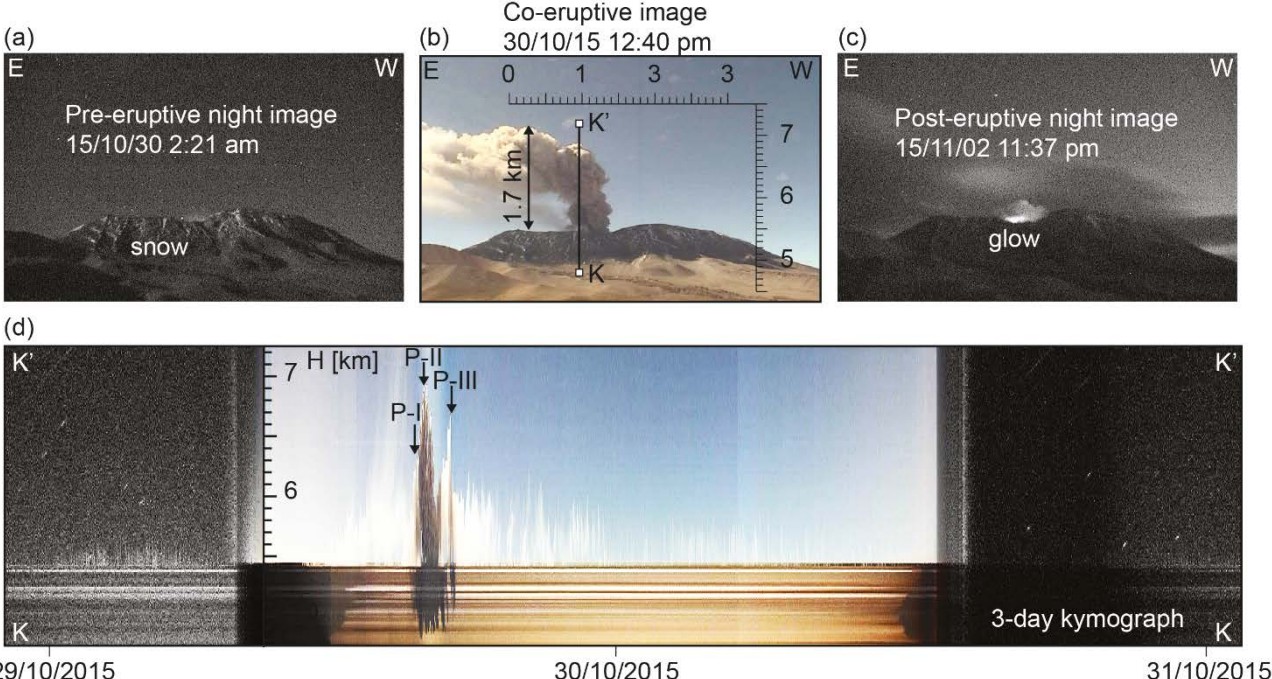

Figure 5: Camera data analysis allowing the eruption duration and height to be estimated from a camera viewing from north to south (C1, see Figure 1 for location). a) Pre-eruptive nighttime image (October 29, 2015, at 23:21 UTC) of the snow-covered edifice. Local time is given in the image. b) Co-eruptive daytime image (October 30, 2015, at 15:40 UTC) with a 1.7 km high ash-loaded eruption column. c) Post-eruptive image (November 3, 2015, at 2:37 UTC) showing increased glow. (d) Three-day kymograph extracted from 2880 images along vertical profile K-K' (see vertical line in Figure 5b); the height scale (H) is shown in kilometres above sea level. P-I, P-II and P-III are the three phases exhibited by the eruptive plume.

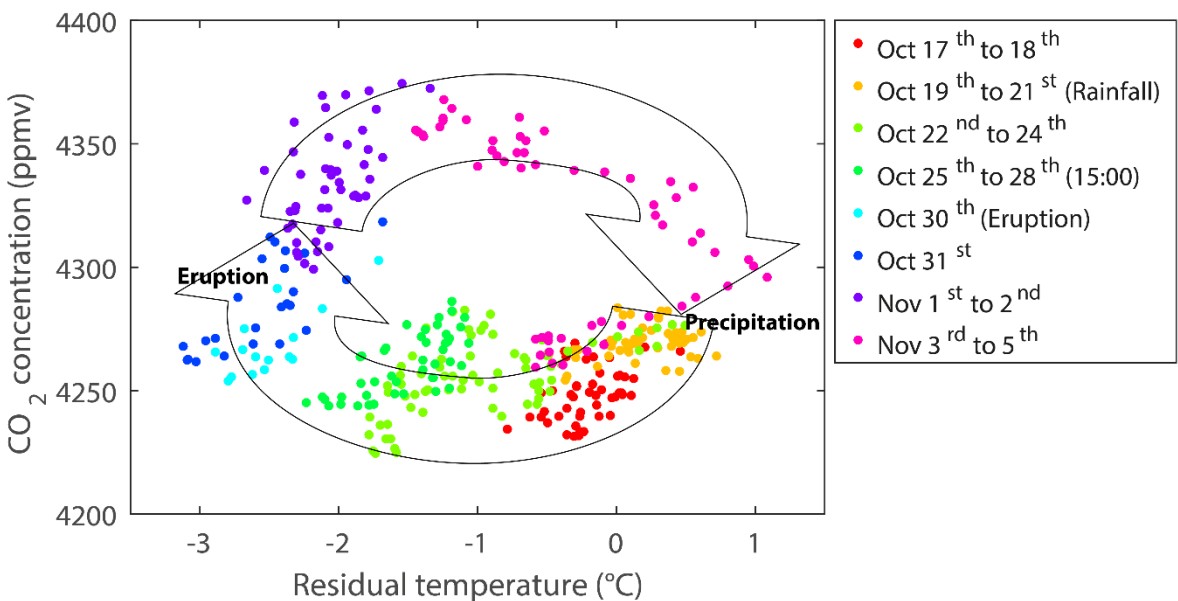

**Figure 6: Evolution plot of the CO₂ mixing ratio versus the gas temperature measured at a Lascar fumarole from October 17 to November 5, 2015. The distinctive coloured dots represent key stages in the evolution of these parameters characterized by a clockwise behaviour. This cyclic behaviour was induced by precipitation event P3 (October 19-21; *orange dots*), causing 1) the gas outlet temperature to decrease following the rainfall event and 2) the CO₂ mixing ratio and outlet temperature to gradually rise subsequent to the eruption (October 30; *cyan dots*). The cycle closes during the period of November 3-5 (*magenta dots*), during which the CO₂ concentration and gas outlet temperature returned to their pre-eruptive values.**



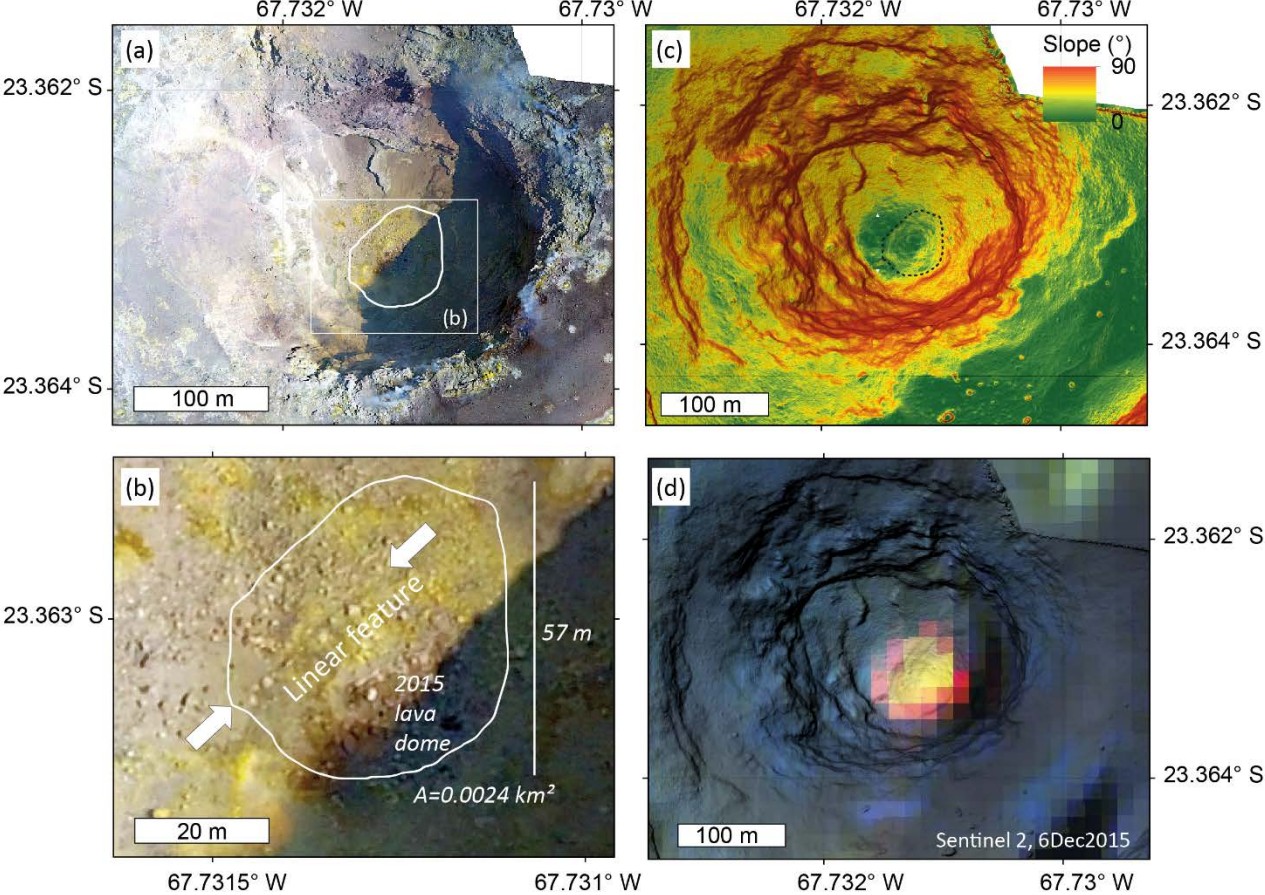

**Figure 7: Drone overflight results. (a) Photomosaic of the crater region showing the 2015 lava dome with a diameter of 57 m. The white line is the approximate perimeter of the dome. (b) Close-up of photomosaic (as indicated in (a)) showing the blocky dome surface and dimensions of the dome. Note the NE-SW-striking linear feature transecting the dome. (c) Digital elevation shaded relief model overlaid by a slope map whose values represent the slope in degrees; the dome is delineated in the centre of the crater. (d) Sentinel-2 thermal anomaly pixels acquired 37 days after the eruption overlain on a shaded relief map to illustrate the location of**
**the dome in the deep Lascar crater.**

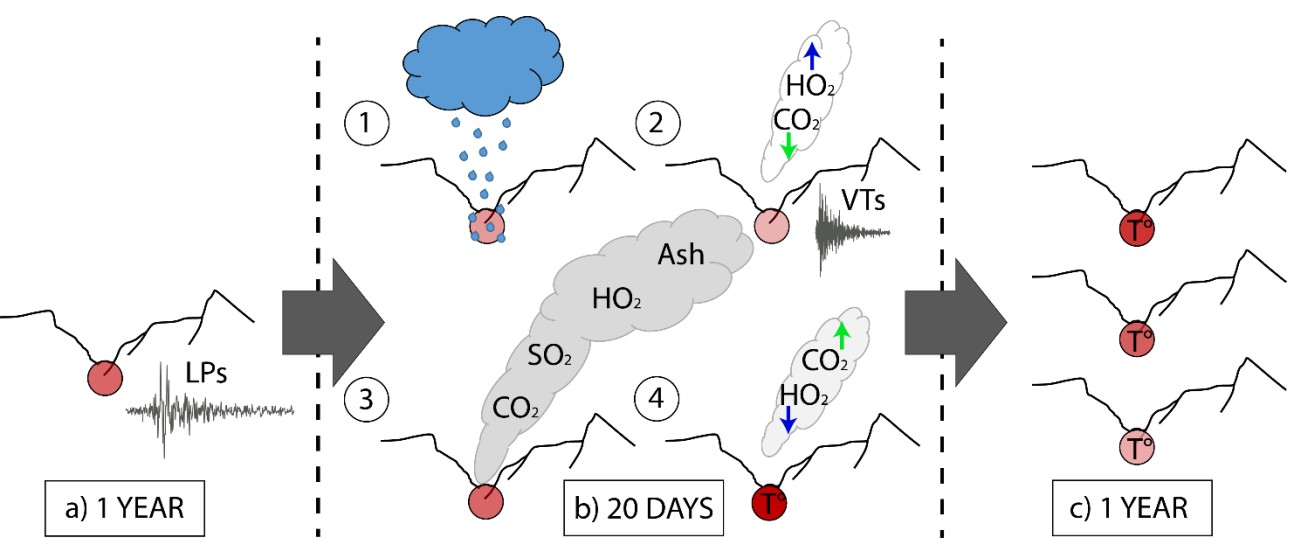

**Figure 8: Schematic interpretation of the pre-, co-, and post-eruptive processes associated with the October 30, 2015, phreatic eruption of Lascar volcano. a) The pre-eruptive phase started approximately 1 year before the eruption, showing a sustained increase in LP activity. b) Ten days before the eruption, a 20-day phase initiated after being triggered by precipitation on October 20th. This phase developed in four stages: 1) An unusually strong precipitation event occurs. 2) Meteoric water percolates into the crater, changes the water content and $CO_2$ concentration in the fumarole and reactivates VT activity. 3) The volcano is pushed into a steam-driven explosion, producing a 1.7 km high eruption column above the crater that is composed mainly of water, ash, $CO_2$ and $SO_2$, causing the dome to fracture. 4) Post-eruptive degassing transpires with a hotter fumarole richer in $CO_2$ with less water as well as a thermal anomaly evidenced in the active crater. c) The post-eruptive permanent thermal anomaly slowly decreased in temperature for 1 year.**