# Peer review of "Processes culminating in the 2015 phreatic explosion at Lascar volcano, Chile, monitored by multiparametric data"

_Natural Hazards and Earth System Sciences, 2019_

## Referee Comment (RC1) · Felipe Aguilera Barraza (Referee) · 25 Jul 2019

The ms is a very interesting document where several data from a number of techniques showing its behavior pre, during and after 2015 Lascar's eruption, become the first very well documented eruption from that volcano, becoming a very important contribution to the knowledge of this volcano and Central Andes volcanoes eruptive style.

However, I have to do mention about several problems in the text, specially in the quality of some data and how is interpreted. here I mention where are the main problems, but in the attached pdf appear detailed comments about each topic.

1. A questionable interpretation of data coming from seismic station, considering that only 1 station was available during the eruption. 2. A major problem is the interpretation of the existence of a lava dome, where only a solidified conduit is present, and mostly of the model is related with the presence of a dome, becoming a doubtable interpretations. 3. Some problems in the techniques can be detected, specially in the case of processing of SO2 data from permanent mini-DOAS station. 4. A better interpretation between rain or snow fall water interaction with hydrothermal-magmatic fluids is needed.

I recommend major corrections, based in a better interpretation of processes, fluid interaction, distribution of hydrothermal-magmatic system, current active crater morphology and conduit details.

Please also note the supplement to this comment:
https://www.nat-hazards-earth-syst-sci-discuss.net/nhess-2019-189/nhess-2019-189-RC1-supplement.pdf

**Supplement:**

[revised manuscript text omitted]

---

## Referee Comment (RC2) · Anonymous Referee #2 · 8 Oct 2019

I found the manuscript very interesting. The authors report a multidisciplinary data set characterizing the phreatic explosion at Lascar occurred in 2015. They observe long-term changes in LP seismic activity preceding the eruption, with a rapid increase in the LP activity about one year before the eruption and a drop in the LP activity 3-6 months before. The decrease of the LP activity is accompanied with a decrease of the persistent thermal anomaly observed in the crater floor.

Two heavy snow events are reported few months before the eruption, not leading to detectable changes at the volcano. However, an heavy rain event about ten days before the eruption is considered by the authors a precursory of the explosion.

[Figure]

The correlation between the rain and the phreatic explosion occurred in 2015 is explained as the heating of the percolated water inside the carapaces of a pre-existing and still hot (?) lava dome in the crater zone.

However, a less clear effect of the lava dome is introduced in section "5.4 Conceptual model", were the lava dome has also the effect of blocking the path of the deep fluids and inducing a long-lasting gradual pressure build-up. This last interpretation is associated to the increase of LP events starting one year before the explosion. I think that the authors should better describe the inter-relationship between pressure increase due to arrival of deep fluids, degassing and the observed subsidence of the crater zone.

Considering a possible seasonal effect on the occurrence of the phreatic explosions at Lascar, and the possible role of the rain, it would be interesting to know, if possible, whether previous phreatic explosions, such as that occurred on 18 April 2006 (ie: after the emplacement of the dome) were preceded by events of heavy rain.

In general, I think that the information contained in the paper can help to better understand the processes leading to the conditions for phreatic explosions.
* * *

---

## Author Comment (AC1) · 19 Nov 2019

Please find below the comments raised by the reviewer 1 followed by our reply. We have structured the response according to the following sequence: (1) comments from Referees, (2) author's response, (3) author's changes in manuscript.

Main general comments (G1-G5):

G1. The ms is a very interesting document where several data from a number of techniques showing its behavior pre, during and after 2015 Lascar's eruption, become the first very well documented eruption from that volcano, becoming a very important

contribution to the knowledge of this volcano and Central Andes volcanoes eruptive style. However, I have to do mention about several problems in the text, especially in the quality of some data and how is interpreted. Here I mention where are the main problems, but in the attached pdf appear detailed comments about each topic.

Reply G1: We appreciate the comments by the reviewer and have addressed all suggestions for improvement of the manuscript. We propose to provide a revised version which is more clear in language, reduces the speculation with respect to interpretation of processes (e.g. about the "dome") and is more careful with the conceptual model provided. Therefore we feel the paper will greatly improve.

G2. A questionable interpretation of data coming from seismic station, considering that only 1 station was available during the eruption.

Reply G2: We agree that only 1 station is weak to determine locations of seismic events, and therefore clarified when and which stations operated. We now clarify that the long-term data presented included a much more complete network, allowing localization (published in Gaete et al., 2019, GJI) and cumulative number and classification (this work). Therefore in the revised version we propose to rewrite the description of data recorded by QUE and provide the reason for relying on them.

G3. A major problem is the interpretation of the existence of a lava dome, where only a solidified conduit is present, and mostly of the model is related with the presence of a dome, becoming a doubtable interpretations.

Reply G3: We accept this comment and agree that the interpretation of the existence of a lava dome was problematic. In the revised version we therefore propose to remove this speculation, and only debate in the discussion chapter possible resemblance to rubble falling back into a crater and to a dome-shaped structure. We lay down arguments for either interpretation. In the rest of the paper there will be no mention of the lava dome anymore. Furthermore, we will rename the discussion section 5.1 to 'Water infiltration into the shallow hydrothermal system of Lascar'.
G4. Some problems in the techniques can be detected, especially in the case of processing of SO2 data from permanent mini-DOAS station.

Reply G4: We thank the reviewer for pointing out these problems in the following specific comments section. Specific answers thus can be found below (Replies 8, 9, 10, 13, 14-4, and 15).

G5. A better interpretation between rain or snow fall water interaction with hydrothermal-magmatic fluids is needed. I recommend major corrections, based in a better interpretation of processes, fluid interaction, distribution of hydrothermal-magmatic system, current active crater morphology and conduit details.

Reply G5: We agree and follow the reviewer's suggestion: In addition to what is already written in Lines 553-562, we propose to provide in the section 5.1 1) a better interpretation of interaction between meteoric and hydrothermal-magmatic fluids (Reply 14-2), and 2) a more detailed description of the hydrothermal-magmatic system, and crater morphology. Corresponding changes in the manuscript will be detailed below (Reply 14-2).

Specific comments (S1-S16):

Abstract:

S1. Line 28: Do not mention as a dome, because there is no a dome, the last evidence of a dome was between December 1993 and the first months of 1994. This should be related to morphological features of the active crater, but not related with a dome.

Reply S1: We agree to avoid the term "dome" here, and propose appropriate changes in the following lines:

Lines 27-29: An increased thermal anomaly inside the active crater observed through Sentinel-2 images and drone overflights performed after the steam-driven explosion revealed the presence of a fracture  ${\sim}50$  metres in diameter truncating the floor of the active crater, which coincides well with the location of the thermal anomaly.

NHESSD
S2. Line 30: Similar than before, in the crater floor is possible to be related to a remanent dome, but is not a dome properly. Matthews et al 1997 mentioned clearly how dome collapsed and the causes producing the collapse. Crater floor must be treated as the highest part of the conduit, independently of what is present.

Reply S2: We agree to this comment and will remove the mentioning of a dome here and rewrite this sentence. We propose to apply appropriate changes in the following lines:

Lines 29-32: Altogether, these observations lead us to infer that a conduit blocking produces sealing of degassing path reduced the thermal anomaly and inhibited gas exhalation. We conjecture that the vaporization of meteoric water percolated into the volcanic system triggered a vent-clearing phreatic explosion. We also discussed the eventual role of the seismicity on the long-term build-up of pressure in an eruptive preparatory phase.

Section 2: Study area

S3. Line 71: A very old reference, must be changed for a new one, e.g. Gardeweg et al 2011.

Reply S3: We will consider the reviewer's suggestion and update the reference to Gardeweg et al., 2011 in line 74.

S4. Line 76: Do you mean 19-21 April sub plinian eruption?

Reply S4: We propose to include the exact date of the eruption in the revised manuscript and to specify that this eruption belongs to the activity cycle which started in January 1993. Appropriate changes will be made in the following lines:

Lines 76-77: The most recent large eruption, classified as having a volcanic explosivity index (VEI) of 4, occurred in 19-20 April, as the climax of the 7-months activity cycle which started in January 1993

NHESSD
S5. Line 76: A better reference is Matthews et al 1997.

Reply S5: We will include this in the reference list in line 80.

S6. Line 85: 17 km west from Lascar, as was mentioned before.

Reply S6: We thank the reviewer for pointing that out. We will correct this information.

Data and analysis method:

S7. Line 224: the southern rim crater is located at 5,502 m a.s.l.

Reply S7: We have checked the altitude and values are in the range  ${\sim}5{,}470$  to  ${\sim}5{,}510$  m a.s.l., therefore in the revised version we will change it to 5,502 m a.s.l. .

**Results:**

S8. Line 245: Cumulative VT/LP means that is the sum of VT and LP events? Or is a ratio VT/LP events. In the first option, the accumulation of seismic events is quite obvious, and sudden increases could be important, but over a period, always will exist an accumulation of events. In the second option, a lower values must be seen, due increasing of LP events. I recommend to show in the graphic VT and LP events.

Reply S8: We agree that our description was unclear and therefore we will make appropriate changes. By the annotation 'cumulative VT/LP' we referred to the cumulative sum of VTs and LPs separately. In the revised version we will modify the graph labelling in Figure 2a and improve the text to make this point more evident and to avoid any future misunderstanding. The proposed changes will be made as follows:

Lines 244-245: In total, 1654 LP (purple dots in Figure 2a) events and 47 VT (green stars in Figure 2a) events were identified during this observation period.

S9. Line 323: Decreasing of the anomaly not necessary represents a temperature decreasing, anomalies can decrease due the number of anomalous pixels decrease, which could be related to decreasing of degassing, sealing of degassing paths, slow
degassing flux, among others.

Reply S9: We thank the reviewer for that hint. We will refrain from talking about a temperature decrease.

Lines 322-325: The dimension and strength of the thermal anomaly slowly declined during 2016 as observed in the Sentinel-2 (Figure 3e-h).

S10. There are two important issues to be consider (line 342):

S10-1. Is very important to highlight that seismic data is only from one seismometer, and must be taken in account in this section, and the discussion, sadly, only was seismometer was available during the crucial eruptive month, and the account of earthquakes can be underestimate, but not give a clear idea about the relations between the eruption and seismic patterns.

Reply S10-1: In the revised manuscript we will consider this, as already stated in the replies above. We observed in the seismic catalogue that the low number of events in October followed the trend that was already observed with the complete network in the previous four months. Additionally, the event detection is performed in real-time and based on a standard protocol that considers the spectral content, signal duration and harmonic signatures performed by expert analysts through visual inspection and classification. Furthermore, the site of the operative station (QUE) provides clear signal records with a high signal-to-noise ratio. Overall and considering the previous reasons, we do not consider that our results show an underestimation in the number of detected events. We propose changes as follows:

Lines 135-137: Four of these five stations were used for the long-term compilation employed for the seismic evolution study from July 2014 to December 2015, and their locations are illustrated in Figure 1. However, as three of these stations stopped functioning before October 2015, only one seismic station was operational throughout the month of the eruption and was useful for assessing the timing and characteristics of
the explosion and simple event classification in October 2015 (station QUE; Figure 1).

Insertion in Line 143: Despite QUE was the only operational station during October 2015, it provided clear signal records with a sufficiently high signal-to-noise ratio (Gaete et al., 2019) to distinguish VT and LP signals with reference to their characteristic spectral contents.

S10-2. I am not absolutely agree about the lava dome. There are no evidences about the presence of a lava dome. The last evidence of a remanent of a lava dome was during the initial months of 1994. The last dome appear 26th April 1993, just 5 days after finished the subplinian eruption. The eruption of December 1993 destroyed almost completely that dome, and the remanent started to disappear with the next several eruptions and because of subsidence. The subsidence is normally produced by decreasing of vesicularity in the base of the dome and its related conduit, which is located in the volcanic conduit. The vesicularity appear due of constant degassing, and the combination of vesicularity and high density of lava dome, produce the collapse of the lava dome body, "returning to the conduit". Those processes are relatively fast, and lasted few months. At the end of 1994, there is no evidences of the presence of a lava dome. After that, several vulcanian and phreatic eruptions have occurred, producing several subsidence periods of the crater floor (more details in GVN, Lascar section; Menard et al 2014). These processes produce a conduit that progressively subside and crater walls collapse. Consequently, is not possible to talk about a dome, must be treated as a conduit roof of crater floor.

Reply S10-2: As already stated above, we agree to be more careful with this terminology and also will adjust our conceptual model. However, we note that the transition between a conduit roof and a lava protrusion are not sharp, therefore this discussion will remain open. To consider this, in the revised version we therefore will improve this description. Changes will be included as follows:

Lines 334-342: Our UAV overflight performed on November 27, 2017, revealed the
presence of a circular feature located at the base of the deep crater floor with a diameter of ~57 m partly covered by rock fall deposits from the crater walls (Figure 7a,c). The circular feature on the crater floor may represent the surface expression of the underlying conduit and/or the remnants of a dome-like protrusion of magma, with a mount-like outline, morphology and blocky appearance at the surface. We compared the location of the circular feature to a thermal anomaly map acquired during the 2015 eruption, and good agreement was observed between the region covered by blocky material and the thermal anomaly region (Figure 7a,b). Close-up views enabled by high-resolution drone photogrammetry further revealed the presence of a linear feature striking NE-SW dissecting this structure but not dissecting the apparently younger rock fall deposits (Figure 7d). The explosive dissection of crater floors and lava domes by linear features has commonly been observed elsewhere following steam-driven explosions (Darmawan et al., 2018a; Walter et al., 2015). Therefore, we speculate that a linear NE-SW-striking fracture developed during the 2015 steam-driven explosion.

Discussion:

S11. Line 351: There is no evidence of magma extrusion since April 1993.

Reply S11: We have considered the reviewer comment, according to observations about the absence of a lava dome, so we will now remove this statement.

Lines 350-352: We noticed that this decline in seismic activity was accompanied by a reduction in the persistent high-temperature anomaly located inside the active crater (Figure 3a-c), which likely was associated with a general decline in fumarole activity.

S12. Line 353: Those authors suggest that decreasing of thermal anomalies are related to: i) Lava dome collapse; ii) sealing of degassing paths, decreasing consequently the bulk degassing, where sealing could be related to the collapse of crater walls. Both cases must be treated differently, due to the origin of the anomalies and the related processes. A more recent work related to Lascar volcano can be found in González et al 2015 (JVGR).
Reply S12: As we declined the speculation about a likely existence of a lava dome (although the shape is "dome-like"), we suggest the inhibition of the thermal anomaly as effect of the crater floor subsidence. Thus, we have now considered that the sealing of the degassing path can be a more plausible option due to a crater subsidence and removed the Oppenheimer et al. 1993 reference and instead included the Gonzalez 2015 reference, as suggested by the reviewer. Instead, we will include the following sentence.

Lines 352-353: Similar decreases in the area and intensity of hot spots have previously been observed preceding, e.g., the eruptions occurred in the periods 1992-1995 and 2000-2004 (See Table 1), which likely have been associated to a sealing of the degassing path probably due to crater subsidence (González et al., 2015; Wooster and Rothery, 1997).

S13. Line 365: One of the biggest questions that appears after to read this and see the data is: Why to consider mid-September to mid-December period, and not mid-January to mid-March? Strongest rainfall occurs during summer season related to "Altiplanic winter", and the period suggested is more related to snowfall. Then, a more dramatic influence of snow could be demonstrated than rain. For this, a better statistical work with eruptions record and seasons must be done

Reply S13: We agree that this discussion paragraph was cloudy. First of all, the used weather sensor is not allowing to speculate on the type of precipitation (snow, rain, hail). To this aim we could use our camera observation, which we now clarified in the method section. For this reason, we carefully corrected the terminology used (precipitation, rain fall, snow fall, hail), which is now consistently described as precipitation events throughout the whole manuscript. We agree that for quantification of the effect of precipitation on volcanic activity a better station network is needed, also allowing to distinguish between rain and snow and hail. We considered mid-September to mid-December because the eruption record shows an eruption recurrence of more than 50% in this period. A more detailed statistical analysis was beyond the scope of this
study. Finally, the limitations regarding to a proper way for precipitation measurements open the discussion about the importance of including weather stations in the monitoring network of phreatic eruptions. This will be added to section 3.6 Weather data, section 4.1 Gradual changes prior to the eruption, and section 5.2 Limitation of the used methods, respectively, as follows:

Lines 237-239 (section 3.6): We considered the intensity and accumulated amount of precipitation measured in a one-minute running average of rain and hail derived from samples acquired every 10 seconds. The rainfall is measured as cumulative accumulation of water on a 60 cm2 area with a range from 0 to 200 mm/h, whereas hail as the cumulative amount of hits against collecting surface. This instrument is not designed to measure snowfall. The data were compared with the other observations to identify a rare precipitation event shortly before the 2015 explosion.

Lines 253-255 (section 4.1): P1, P2 and P3 were characterized mainly by snowfall as was observed by our IP cameras. Nevertheless, our weather station detected considerable amounts of precipitation during events P1 and P2 which occurred in the middle (March 2015) and end (August 2015) of the increasing LP activity phase, respectively.

Insertion in Line 481 (section 5.2): In this elevated Altiplano zone, storms and in general bad weather conditions are more often characterized rather by snowfall than rainfall or hail, as was observed by camera during the three precipitation events that are covered by this study. However, distinguishing between the types of precipitation (snow, hail, rain) in such field conditions is challenging, as strong winds, dry atmosphere and sublimation instead of melting, besides other complexities, may lead to some precipitation events that have an effect on the volcano, while others do not. Therefore values registered by our instrument can be an underestimation of the real amount of accumulated precipitation, but also do not necessarily reflect changes in soil moisture and water penetration. To understand the occurrence of phreatic eruptions in the Altiplano zone, future monitoring networks should include hydrometeor and soil moisture stations capable of distinguishing between types of precipitation and water penetration. NHESSD
S14. Several questions and comments appear after to read this section (5.1):

S14-1. The relation of the eruptive process-rainfall-lava dome is doubtable regarding the no presence of a lava dome, and consequently, the process must be treated in other way.

Reply S14-1: We agree with the reviewer about the need of avoiding the use of 'lava dome' and we propose to reword the section in large parts (see also next point). We will make according changes in Figures 7 and 8 and respective figure captions.

Caption of Figure 7: Drone overflight results. (a) Photomosaic of the crater region showing the 2015 central elevation of the crater floor with a diameter of 57 m. The white line is the approximate perimeter of the thermal anomaly. (b) Close-up of photomosaic (as indicated in (a)) showing the blocky central elevation of the crater floor which corresponds to the dimensions of the thermal anomaly. Note the NE-SW-striking linear feature transecting the central elevation. (c) Digital elevation shaded relief model overlaid by a slope map whose values represent the slope in degrees; the central elevation is delineated in the centre of the crater. (d) Sentinel-2 thermal anomaly pixels acquired 37 days after the eruption overlain on a shaded relief map to illustrate the location of the central elevation of the central elevation of the central elevation of the central elevation of the central elevation.

Caption of Figure 8: Schematic interpretation of the pre-, co-, and post-eruptive processes associated with the October 30, 2015, phreatic eruption of Lascar volcano. Light blue stripes represent the approximate location of the hydrothermal system of Lascar a) The pre-eruptive phase started approximately 1 year before the eruption, showing a sustained increase in LP activity. b) Ten days before the eruption, a 20-day phase initiated after being triggered by precipitation on October 20th. This phase developed in four stages: 1) An unusually strong precipitation event occurs. 2) Meteoric water percolates into the crater, changes the water content and CO2 concentration in the fumarole and reactivates VT activity. 3) The volcano is pushed into a steamdriven explosion, producing a 1.7 km high eruption column above the crater that is Interactive comment

composed mainly of water, ash, CO2 and SO2, causing the crater floor to fracture. 4) Post-eruptive degassing transpires with a hotter fumarole richer in CO2 with less water as well as a thermal anomaly evidenced in the active crater. c) The post-eruptive permanent thermal anomaly slowly decreased in temperature for 1 year.

S14-2. According to the arguments, one of the good explanation is about the relation between the rain (snow) fall and its interaction with the hydrothermal system. Sadly, is poorly discussed. There are a couple of works demonstrating that hydrothermal system has been identified in Lascar volcano (Aguilera 2010; Tassi et al 2009; Menard et al 2014). Tassi et al 2009 has a good model about the magmatic-hydrothermal systems distribution. That model suggests a peripheral hydrothermal system, whereas in the zone related to crater floor, gases show a predominance of a magmatic system. Following that model, must be explored the interaction between magmatic gases and rainfall. However, dynamics of hydrothermal-magmatic systems is sometimes very fast, and interaction between hydrothermal gases and rainfall must be better discussed.

Reply S14-2: Our data suggests that the eruption released gases coming directly from the magmatic system. This is supported by the SO2 peak during the eruption and the thermal anomaly increase and a glow after the eruption. On the other hand, direct measurements of CO2 in a peripheral fumarole show variations in temperature and CO2 content that we associated with the precipitation event. Due to the location of our geochemical station is evident that this reflects the effect of meteoric input as discussed by Tassi et al 2009; Menard et al 2014. We will add the following paragraph in section 5.1 to this regarding:

Insertion in Line 362: The hydrothermal system of Lascar has been extensively studied before (Menard et al., 2014; Tassi et al., 2009). Gas emissions occurring at the crater floor have been previously characterized by discharge of fluids fed by a deep magmatic source. Lascar host an extended hydrothermal system feeding the fumaroles located on the inner crater walls and the upper rim of crater (Gonzalez et al., 2015). Gas emissions from these fumaroles show an increasing hydrothermal chemical signature

NHESSD
with increasing distance from the magmatic body. Therefore the hydrothermal system encompasses a central system (magma-dominated), and a peripheral system (meteorically dominated), which is susceptible to interact with meteoric water added into the system (Tassi et al., 2009). Our study supports a link between these two systems, as we show evidence that fumarole measurements taken on the outer crater rim are displaying changes related to both the precipitation event and steam driven explosion which are explained in detail in the section 5.4. Therefore the outer and inner hydrothermal system appear to be dynamically linked, either to eruption occurrences, or to precipitation events, or to both as our study suggests (see also Figure 6). Important implication arise, as monitoring the outer system, which is easy to access, may even allow obtaining an indirect glimpse of the inner hydrothermal system.

S14-3. Is not clear the relation between thermal anomalies and pre-post eruptive periods. Your data show a very similar behavior as showed by Oppenheimer et al 1993, Wooster and Rothery 1997, Gonzalez et al 2015, in seems that sealing of degassing paths is a plausible explanation, related to i) collapse of crater walls and/or ii) sealing by presence of hydrothermal carapace. Higher SO2 flux after eruption is compatible with this model.

Reply S14-3: We carefully read the suggested references (by Oppenheimer et al 1993, Wooster and Rothery 1997, Gonzalez et al 2015) and agree that there is a very strong similarity with respect to the temporal evolution of thermal anomalies observed prior to and following eruptions. According to our satellite observations, the location of the thermal anomaly is located directly over the active crater. The decrease in size and intensity of the thermal anomaly would be associated to the sealing of the area with major contribution of the magmatic gasses and less influenced by meteoric input, in accordance with the interaction model suggested by Tassi et al. 2009. Thus, the post-eruptive increase of SO2 flux would agree with the opening or unblocking of the volcanic conduit due to the eruption. We thus propose to make following changes:

Lines 350-354: We noticed that this decline in seismic activity was accompanied by a
reduction in the persistent high-temperature anomaly located inside the active crater (Figure 3a-c), which likely was associated with a general decline in fumarole activity. Similar decreases in the area and intensity of hot spots have previously been observed preceding and following, e.g., the eruptions that occurred in the periods 1992-1995 and 2000-2004 (See Table 1), which likely have been associated to a sealing of the degassing path probably due to crater subsidence (González et al., 2015; Wooster and Rothery, 1997).

S14-4. I have a lot of doubts about seismic data, using only one seismometer cannot give a good idea about VT events and the eruptive period. VT can be related to deeper processes and not necessary to shallow conduit-related events. With at least 4 stations will be possible to have a good VT location, and then try to correlate it. In fact, shallow conduit-related events should be more related to LP and tremors events, and LP events in the more proximal days from the eruption are not showed or explained. Additionally, is very hard to talk about precursors with only one station

Reply S14-4: We agree that the presentation of the seismic data was lacking necessary details. In the revised version we clarify that the preparatory phase observed from LP activity was based on a large network, allowing event classification (LP and VT events) and also localization. Details on this were recently published in a separate paper (Gaete et al., 2019), which we will refer to in relevant places in the revised manuscript. The 1-month period prior to the eruption indeed was monitored only by one single station, but based on the knowledge obtained from the earlier complete network, also this single station allowed classification. Localization was not possible with this single station, however. These station details we will clarify in the revised manuscript as described in Reply S10-1.

S15. Line 480: A mention must be done about SO2 retrieval process, OVDAS monthly reports show unrealistic high SO2 fluxes, and probably is related to atmospheric model used for wind speed. Processing must be done carefully, and the model for wind speed used is critical when SO2 fluxes ares estimated. A more accurate description of the
SO2 processing should be done, and then put in context which type and quality of data is presented.

Reply S15: We appreciate the particular interest of the reviewer in the SO2-flux retrieval process. The SO2 retrieval process itself was sufficiently described in section 3.3 of the submitted manuscript, however we agree that a short description of the error bars and quality of the data selected for evaluation would be valuable to add, which we will do in section 5.2 of the revised manuscript. We decided not to further expand on the topic of different evaluation strategies in the manuscript, because the scanning DOAS method and SO2-flux data are not the main focus of the paper. For sake of transparency instead we will provide detailed insight into our evaluation strategy in our answer below, whereas in the previously submitted manuscript we already made sure to properly cite the most relevant studies that can be followed to describe the SO2 retrieval process.

**Evaluation strategy used for SO2-flux retrieval**

The slight discrepancy between time averaged values reported here and monthly values reported in the OVDAS reports is the result of a different evaluation strategy that we applied for the retrieval of the SO2-fluxes presented in this paper, rather than a result of the choice of modeled wind speeds as a source for estimation of gas plume transport speeds as speculated by the reviewer. The latter was used for both, retrieval of values in the manuscript and the values in the OVDAS reports, and its uncertainties will be detailed further below. In this work a different processing strategy was chosen with respect to the one used at the observatory, in order to obtain a more complete time series, however, at the expense of a larger variability of data quality. The retrieval process utilized in this paper does not apply the thresholds used by OVDAS, which requires time consuming manually conducted tests to generate a prior knowledge of plume movement between consecutive scans and its effects on spatial gas distribution in gas plume cross-sections. Such a procedure thus is not yet applicable for near real-time monitoring purposes with available software tools. The retrieval process used
at the volcano observatory in comparison is much more conservative with respect to uncertainties in data quality. The data included in the OVDAS monthly reports fulfill the following three requirements to be validated: 1) The volcanic plume covered by the instrument must be bigger than 70% of the plume width estimated by the evaluation software. 2) The shape of the plume cross-section must be more or less symmetric with respect to the centre of mass in the cross-section. 3) For providing reliable measurements good weather conditions are required with clear or mostly clear sky which is checked utilizing imagery from IP cameras. The downside of this selective evaluation, which exclusively picks the most reliable data, is a significantly reduced data coverage during periods with unfavorable weather conditions, which in turn may eventually lead to a statistics which is shifted in favor and towards the high values of the actually possible measurement range provided by such an instrument, i.e. a slight overestimation with respect to the complete statistics, which also includes more uncertain measurements that are prone to underestimation of the gas flux. We generally do not agree with the reviewer, that the SO2 -fluxes provided by the monthly OVDAS reports are unrealistically high. The values reported in the monthly reports typically lie well within the range of values that have been reported elsewhere in the literature (Mather et al., Menard et al., 2014).

Uncertainties of modelled wind speeds

We agree with the reviewer that the quality of modeled wind speeds can be highly variable in time and moreover differs strongly from location to location. Modeled wind speeds thus require to be tested against field observations in order to ensure data quality and to identify tendencies of deviation from real weather conditions. At Lascar volcano the wind speeds estimated by the GFS hindcasts have the tendency to be rather lower than those retrieved from weather stations in the area. The wind speeds that we extracted for the location and altitude of our weather station at the base of the volcano during the period of this study were at average 0.32 m/sec lower than those measured by the weather station. The median of that deviation between model and
station was -0.44 m/sec, and typical deviations were not larger than  $\pm 2.9$  m/sec (standard deviation). Average wind speed at about 200 m above the summit of Lascar was 14.4 m/sec during respective period. Typical errors were thus not larger than  $\pm 20.4\%$ , and the resulting average error of the modeled wind speed is -4.6%, which is why the corresponding SO2-flux calculated from modeled values rather likely is systematically underestimated. Considering that errors in wind speeds usually constitute the largest contribution to the total error of the scanning DOAS method, these errors however are comparatively low and therefore the GFS model is a reliable source for gas plume transport speed estimates, particularly at this volcano.

Insertion in Line 480: Altogether, these frequently occurring unfavourable weather conditions led to a variable data quality, resulting in standard errors ranging between  $\pm 30$  and  $\pm 50\%$ .

S16. Line 487: According to the fumarole characteristics and its temperature, seems that is a hydrothermal related fumarole. According to the discussion, this fumarole could be a very good fumarole, regarding a possible interaction between hydrothemal fluids and rainfall water.

Reply S16: We agree that it may be stressed even further that the susceptibility of fumarole characteristics to changes in weather was indeed very beneficial for our study and worked out particularly well at our hydrothermally dominated low-temperature fumarole on the crater rim. Changes will be done as follows.

Lines 489-492: Furthermore, the CO2 mixing ratio and temperature of a fumarole are also strongly susceptible to changes in the weather, which is why the interpretation of these variables is often complicated when observed trends cannot explicitly be attributed to weather conditions or volcanic activity. Monitored together with relevant weather variables, however, measurements of CO2 mixing ratio and temperature in hydrothermally dominated fumaroles are extremely well suited for studying the influence of weather conditions on volcanic activity. This was particularly true for our hydrother-
mally dominated low-temperature fumarole on the crater rim as was demonstrated in this work.

---

## Author Comment (AC2) · 19 Nov 2019

Please find below the comments raised by the reviewer 2 followed by our reply. We have structured the response according to the following sequence: (1) comments from Referees, (2) author's response , (3) author's changes in manuscript.

Main general comments (G1-G3): G1. I found the manuscript very interesting. The authors report a multidisciplinary dataset characterizing the phreatic explosion at Lascar occurred in 2015. They observe long-term changes in LP seismic activity preceding the eruption, with a rapid increase in the LP activity about one year before the eruption and a drop in the LP activity 3-6months before. The decrease of the LP activity is ac-

companied with a decrease of the persistent thermal anomaly observed in the crater floor. Two heavy snow events are reported few months before the eruption, not leading to detectable changes at the volcano. However, a heavy rain event about ten days before the eruption is considered by the authors a precursory of the explosion.

Reply G1: We appreciate these comments

G2. The correlation between the rain and the phreatic explosion occurred in 2015 is explained as the heating of the percolated water inside the carapaces of a pre-existing and still hot (?) lava dome in the crater zone. However, a less clear effect of the lava dome is introduced in section "5.4 Conceptual model", were the lava dome has also the effect of blocking the path of the deep fluid sand inducing a long-lasting gradual pressure build-up. 2. This last interpretation is associated to the increase of LP events starting one year before the explosion. I think that the authors should better describe the inter-relationship between pressure increase due to arrival of deep fluids, degassing and the observed subsidence of the crater zone.

Reply G2: We will include the following more detailed explanation already in the introductory paragraph of the discussion section (5) in order to make clearer the interaction between the pressure and the blocking of the degassing path:

Lines 344-354: The steam-driven explosive eruption of Lascar on October 30, 2015, was the first that was densely monitored. The eruption was studied by utilizing different data streams, the results of which suggest that (i) no magma movements within a shallow magma reservoir were identifiable immediately prior to the explosion though significant changes in degassing activity were observed and (ii) the spontaneous steam-driven explosion was directly associated with a brief degassing pulse and the development of a fractured dome-shaped feature on the crater floor. We ascertained that the volcano was in an elevated stage of activity, as the steam explosion was preceded by ∼1 year of enhanced LP seismic activity thus favouring a potential gradual pressure build-up within the shallow volcanic system. However, as the seismic activity gradually

declined approximately 4 months prior to the explosion (Figure 2a), a direct and causal relationship is debatable. Nevertheless, similar long-term trends in LP activity were observed prior to eruptions of Mt. Etna, which modulation was associated to replenishment with gas-rich magma (Patanè et al., 2008). If that was the case also at Lascar, this would imply a considerable input of deep gas/fluid into the system which release may eventually have been obstructed by reduction of permeability of the degassing path in response to the precipitation (Heap et al., 2019), increasing the pressure in the volcanic system. We noticed that this decline in seismic activity was accompanied by a reduction in the persistent high-temperature anomaly located inside the active crater (Figure 3a-c), which likely was associated with a general decline in fumarole activity. Similar decreases in the area and intensity of hot spots have previously been observed preceding, e.g., the eruptions that occurred in the periods 1992-1995 and 2000-2004 (See Table 1), which likely have been associated to a sealing of the degassing path probably due to crater subsidence (González et al., 2015; Wooster and Rothery, 1997). The details of our findings, limitations and interpretations as well as a conceptual model will be discussed in the following.

G3. Considering a possible seasonal effect on the occurrence of the phreatic explosions at Lascar, and the possible role of the rain, it would be interesting to know, if possible, whether previous phreatic explosions, such as that occurred on 18 April 2006 (ie: after the emplacement of the dome) were preceded by events of heavy rain. In general, I think that the information contained in the paper can help to better under-stand the processes leading to the conditions for phreatic explosions.

Reply G3: We appreciate this comment and started investigating, but we note that retrieving such potentially archived data is rather challenging: Unfortunately we neither have precipitation data from 2006, which would enable us to answer this question, nor it is indicated in eruption records, whether the 2006 eruption was preceded by a rainfall event. However we emphasize that based on the eruption history (Table 1), we are able to show that 50% of the documented eruptions occurred in springtime only.

A similar seasonality effect of volcano eruptions was also inferred at Iceland, where most large eruptions occur during spring and summer periods (Albino et al., 2010). To understand the occurrence of phreatic eruptions, future monitoring networks should include hydrometeor stations capable of recording rain, hail and snowfall.

We thus propose to insert the following:

Line 366-368: Likewise, the October 2015 eruption falls within this period and occurred only a few days after a precipitation episode, which possibly led to the observed eruption. A similar seasonality effect of volcano eruptions was also inferred at Iceland, where most large eruptions occur during spring and summer periods (Albino et al., 2010).
* * *